# Barley, an Undervalued Cereal for Poultry Diets: Limitations and Opportunities

**DOI:** 10.3390/ani12192525

**Published:** 2022-09-21

**Authors:** W. Nipuna U. Perera, M. Reza Abdollahi, Faegheh Zaefarian, Timothy J. Wester, Velmurugu Ravindran

**Affiliations:** 1Monogastric Research Center, School of Agriculture and Environment, Massey University, Palmerston North 4442, New Zealand; 2Department of Animal Science, Faculty of Agriculture, University of Peradeniya, Peradeniya 20400, Sri Lanka

**Keywords:** barley, enzymes, feed processing, non-starch polysaccharides, poultry, β-glucan

## Abstract

**Simple Summary:**

With the ever-increasing demand for poultry products, the continuous supply of conventional cereal grains such as maize has become a challenge. Barley has been recognised as a potential alternative feed ingredient that can replace common cereal grains in poultry diets. However, due to several limitations such as the presence of various anti-nutritive factors and the variability in nutrient composition and quality, the use of barley in poultry diets remains comparatively low. The previous findings on the optimum use of barley in poultry diets are also inconsistent primarily due to differences in research methodologies. The importance of using accurate nutrient profiles for specific barley cultivars to formulate barley-based diets is emphasised in this review. Moreover, the need to adapt feed processing conditions suitable to different barley cultivars to increase the inclusion of barley in poultry diets is highlighted in this review.

**Abstract:**

The supply of conventional cereal grains, especially of maize, will be a significant constraint to the future growth of the poultry industry. Various alternative feed ingredients are being tested to replace maize in poultry diets. Barley (*Hordeum vulgare* L.) is one such feed ingredient, the use of which remains limited in poultry diets due to its low metabolisable energy, presence of anti-nutritive, soluble non-starch polysaccharides and consequent inter-cultivar variability. Differences in research methodologies used in published studies have also contributed to the inconsistent findings, preventing a good understanding of the nutritional value of barley for poultry. The importance of using accurate nutrient profiles, specifically metabolisable energy and digestible amino acids, for specific barley cultivars to formulate barley-based diets is emphasised. Nutritionists should also pay close attention to feed processing conditions tailored to the specific barley cultivars to increase the barley inclusion in poultry diets.

## 1. Introduction

It is projected that the global demand for eggs and poultry meat will increase in the future, and such a growth will have a profound effect on demand and cost of feed materials. In consequence, the supply of traditional raw materials, especially of energy sources, cannot be met even with optimistic forecasts. The first strategy available to the industry is to expand the feed resource base by evaluating and using alternative energy sources, and barley (*Hordeum vulgare* L.) is one such underexploited cereal. Despite the interest, only one review on barley is available in the literature [1]. The aim of the current review is to provide a comprehensive discussion of research to date on the feeding value of barley for poultry. It is hoped that this treatise will offer much greater clarity and understanding of the research topic.

## 2. Classification of Barley

Barley, one of the first domesticated crop, has played a role of multipurpose grain as both food and feed throughout the history. It is extensively cultivated, ranking fourth in world cereal production with an annual production of 157 million metric tonnes [2]. Characteristics such as resistance to drought and saline soils [3] and the ability to mature in climates with short growing seasons [4] have encouraged the cultivation of barley over maize and wheat. In addition to the common usage of barley for malting and brewing (90% of total barley production [5]), it is also used as a feed ingredient in animal diets, especially in Europe where there is the highest concentration of barley cultivation [6,7]. According to records on barley use in animal feeds, 40% of the barley is fed to feedlot cattle, 34% to dairy cows, 20% to pigs and 5% to grazing ruminants, and only less than 1% used for poultry [8].

Morphological and physico-chemical characteristics have laid the foundation for classification of barley. Barley cultivars are classified based on factors such as presence or absence of an awn (a bristle-like appendage), number of the seeds on the stalk, presence or absence of the hull, composition of the starch, aleurone colour and growth height. Moreover, barley is classified according to the growing season as spring or winter cultivars. More genetic selection has been performed on spring barley cultivars, which contain greater energy value [9] and higher resistance to extreme environmental conditions compared to the winter cultivars [10]. The classification of barley based on morphological and physio-chemical characteristics has been comprehensively discussed in Jacob and Pescatore [1].

Classification of barley based on the presence or absence of a hull that contributes to the insoluble fibre fraction [4] is of particular interest to poultry nutritionists. Hull-less or naked barley appears similar to hulled barley until maturity and, then the hulls are loosened and detached during harvesting [11]. In addition to hulled and hull-less barley cultivars, dehulled and pearl barley are produced by the processing of barley grain. Dehulled barley, which is often confused with hull-less barley, is formed by removing the hull from hulled barley. Pearl barley is developed from steam processed and polished (also known as abrading or pearling [12]) dehulled barley. The major difference between dehulled and pearl barley is the presence of both bran and germ in dehulled barley, and absence of bran in pearl barley.

## 3. Composition

The composition and properties of barley grain are of interest in nutritional studies for their role in determining the availability of nutrients to humans or animals. The large variations in composition, structure and physico-chemical properties in different barley types can provide the basis for the differing responses observed among experiments. Extensive research on the composition of barley has recognised that the wide diversity is mainly associated with the differences in hull and starch type, which will be considered as the basis of comparison in this review.

### 3.1. Structural Composition

Barley grain is composed of a large endosperm (80% of the cereal grain), an embryo and a mass of maternal tissues. Mature endosperm consists of five types of cells, as aleurone, sub-aleurone, starchy endosperm, embryo-surrounding region, and endosperm transfer cells. Endosperm cells are filled with starch granules embedded in a protein matrix [13] and, therefore, possess a greater nutritional value compared to other parts of barley grain. The embryo is rich in lipids and enzymes while the aleurone layer is rich in soluble protein (about 50%) and is a source of several endogenous enzymes, lipids and vitamins [14]. Endosperm cell walls are thinner than cell walls of other regions in barley grain and, are mainly composed of β-glucans (70%) and smaller amount of arabinoxylans (20% [15]). However, aleurone cell walls are mainly composed of arabinoxylans (67−71%) and smaller amounts of β-glucans (26% [16]).

### 3.2. Chemical Composition

Wide variability in the chemical composition in different barley cultivars has been reported in the literature [9,17,18], and considerable variation was observed even among similar cultivars [19,20]. Minor changes in chemical composition may result in significant changes in nutrient availability, with remarkable effects on the nutritional quality of barley for poultry [13,21]. 

Environmental factors such as geographical location [4,10,18,21], year of harvest [4], rainfall [22], soil conditions and fertilisation [23], temperature during grain fill [24,25] and storage conditions [26] can affect the chemical characteristics of barley. Varying effects caused by environmental factors on chemical composition of barley highlight the need to consider the environmental conditions, when comparing the chemical composition of different barley types. Rodehutscord et al. [27] analysed the composition of different cereal grain genotypes grown in the same site, thereby excluding the influence of location, management and fertilisation on nutrient composition. Barley of different genotypes still substantially differed in their chemical composition and physical characteristics.

#### 3.2.1. Starch

In common with other cereals, starch is the main component in barley (513 to 642 g/kg DM [28]) and serves as the primary source of energy for poultry fed barley diets. Barley starches differ widely in amylose to amylopectin ratios resulting four different barley types as normal, high amylose, waxy and zero amylose waxy barley types (Table 1). The starch in normal barley genotypes consists of 650−840 g/kg amylopectin, and waxy starch consists of 850−1000 g/kg amylopectin [29,30]. Barley types with 1000 g/kg of amylopectin are termed as zero amylose waxy and high amylose barley cultivars contain around 550 g/kg amylopectin [5]. Waxy gene originated from natural mutations affecting the synthesis of amylose [31], was originally found in maize and later incorporated into barley [1].

Even within the same starch type, amylose to amylopectin ratio can vary widely (Table 1). Nevertheless, some studies evaluating the feeding value of different barley types for poultry have only reported the starch type with no information on the amylose to amylopectin ratio [35]. As even a minor change in amylose to amylopectin ratio can affect the utilisation of starch and performance of poultry [36], it is recommended to consider the starch characteristics beyond already established classifications on starch types. In comparison to amylopectin rich starch, high amylose starch is less susceptible to enzymatic degradation by α-amylase in small intestine, highlighting that waxy starch may be more digestible than the normal starch type [37]. However, most information on the effects of structure and integrity of dietary starch granule and changes of amylose: amylopectin ratio on starch digestion are based on starches from wheat and maize [31,36] and conducted in vitro [38,39,40]. Therefore, careful attention should be given when drawing conclusions from those studies for the barley diets especially due to the interference of non-starch polysaccharides (NSP) in barley. Moreover, in most studies with barley, despite of being the major energy component, no attempt was made to identify starch type and to quantify the starch contents, highlighting a major limitation in barley-related nutritional studies.

Starch granules in both wheat and barley are known to have a bimodal size distribution with small (≤10 µm of diameter) spherical B-granules and large (>10 µm of diameter) disc-shaped A-granules [5,41,42]. Li et al. [5] reported a wide range of starch granule sizes (4.0 to 18.8 µm) in barley compared to maize (6.3 to 13.2 µm), and a negative correlation between starch granule diameter and total amylose content. Moreover, the ratio of number of small granules to large granules in barley starches vary widely compared to maize starch and, the proportion of small granules was correlated with total amylose content [5]. In addition to the chemical characteristics that determine the contribution of barley starch to feeding value, functional properties of starch such as granule structure, size, shape, surface area and interactions with other nutrients (proteins and lipids [5]) can affect the accessibility of starch granules by digestive enzymes and thus the rate and extent of starch digestion, as extensively discussed in Svihus et al. [31].

#### 3.2.2. Protein and Amino Acids

In contrast to plant protein sources, cereals contain lower amounts of CP and AA. Nevertheless, owing to the high inclusion of cereal grains in poultry diets, cereal proteins make a substantial contribution (30−35%) to the supply of dietary AA. The CP content of barley can vary between cultivars (Table 2) and cultivation practices, and nitrogen (N) fertilisation can have a huge impact. Nitrogen fertilisation was shown to increase the CP content in different barley types regardless of hull and starch type [23,43]. Moreover, the relative contents of essential AA to CP in barley were decreased with increasing contents of CP [1,10]. Despite standardised growing conditions maintained by growing in the same site and thereby excluding the influence of location, management and fertilisation on nutrient composition, Rodehutscord et al. [27] reported a range of CP (from 108 to 136 g/kg DM; 6.0% coefficient of variation) for eight winter barley types.

Similar to other grains, barley protein is low in lysine, threonine, methionine and histidine. However, compared with maize and wheat proteins, barley protein has more favourable AA composition (Table 2). Moreover, barley has more protein compared to maize, indicating its nutritional potential [27,45]. In barley, maize and wheat, methionine concentration was the lowest followed by histidine and cysteine, while glutamic acid was the highest [27], regardless of the starch or hull type [13]. Maize protein is higher in leucine and lower in lysine concentrations, compared to wheat and barley proteins [27,45]. A negative correlation between starch and protein contents has been observed in different barley types [5,28]. It has been commonly observed that when the content of starch increases, all other main constituents decrease.

The absence of hull was known to influence the protein content [11,47]. However, both lower [34] and higher [13,28] CP contents reported for hull-less barley compared to hulled barley suggests that CP was independent of the hull (Table 2). The lack of attempts to distinguish between different barley types evaluated in some extensive studies [27,46] has narrowed the opportunity to interpret the influence of starch type and hull on CP and AA concentration. However, according to limited literature on AA comparison in different barley types [13,34], the differences in AA composition seems to be influenced by the CP content, rather than the starch type or hull.

#### 3.2.3. Non-Starch Polysaccharides

Non-starch polysaccharides belong to the fibre component in cereal grains, which is mainly from the cell wall structure [48]. Encapsulation of nutrients within endosperm cells and increased intestinal digesta viscosity are two major mechanisms whereby NSP impair digestion and absorption of nutrients in birds fed diets based on viscous grains. Based on the solubility in water, NSP are categorised into two main fractions namely insoluble (I.NSP) and soluble NSP (S.NSP) [49]. In contrast to the relatively constant S.NSP proportion in wheat [49,50], a wide range of S.NSP has been reported in barley (Table 3). 

##### Insoluble Non-Starch Polysaccharides

Insoluble fibre in cell walls creates a cage effect by encapsulating nutrients (starch and protein) in the barley endosperm, and act as a physical barrier restricting the contact with digestive enzymes, and consequently limit the feeding value of barley in poultry diets. It has been demonstrated that the cell walls in the endosperm of barleys with high levels of β-glucans were thicker than in barleys with low levels of β-glucans [16,23]. It can be therefore speculated that waxy and high amylose barley types with a higher content of β-glucan may be more affected by the cage effect than other barley types. A higher level of I.NSP has been reported in hulled barley types compared to hull-less barley types, due to the presence of hulls [28,52], suggesting greater occurrence of I.NSP in the hull compared to the barley kernel. Comparing 12 hulled and six hull-less barley types, Beames et al. [52] reported that hulled and hull-less barley types differed mainly in the I.NSP (11.5−17.3% vs. 6.6−8.7% DM, respectively) and lignin (1.7–4.5% vs. 0.7–1.3% DM, respectively) contents.

It is universally assumed that the I.NSP is a nutrient diluent with little or no effect on nutrient utilisation [53]. Nevertheless, the benefits of incorporating insoluble fibre in poultry diets by assisting gut motility [54], gut development and health [55], digesta transit time [48], nutrient digestion [56] and bird behaviour [57] are being increasingly recognised and extensively discussed in comprehensive reviews [57,58,59]. Consequently, it is now recommended to include low to moderate amounts of coarse I.NSP, such as wood shavings [60,61] and oat hulls [62,63,64], at 20 to 30 g/kg to low fibre broiler diets [58,65] and 50 to 70 g/kg to layer diets [66]. 

Majority of the benefits of I.NSP on enhanced nutrient utilisation and growth performance is a consequence of improved gizzard functionality and nutrient utilisation, and the effect is more pronounced for starch digestion. Svihus [67] observed greater starch digestibility for a barley-based diet (0.96), compared to four wheat types (0.80, 0.76, 0.83 and 0.73), a finding that was attributed to greater gizzard development influenced by I.NSP available in barley diets [56]. A surplus of starch in the digestive tract can result in low starch digestibility in broiler chickens, but a functional gizzard can prevent the over accumulation of starch [56] by regulating the digesta passage rate [57]. However, the positive effects of I.NSP also depend on grain physical characteristics such as particle size and hardness [57].

##### Soluble Non-Starch Polysaccharides

Due to the high level of S.NSP, barley is categorised as a viscous cereal together with rye, wheat, triticale, and oats. Partially soluble mixed linkage (1→3), (1→4)-β-D-glucan and arabinoxylans have been identified as the main NSP present in both wheat and barley compared to maize. While β-glucan is prominent in barley, arabinoxylans are the predominant NSP in wheat. Though both wheat and barley have higher levels of NSP compared to maize, barley NSP mainly consists of the soluble fraction compared to wheat [68]. Soluble NSP form a gel and increase intestinal viscosity, resulting in reduced accessibility of digestive enzymes to nutrients [4,69,70]. Moreover, increased digesta viscosity can modify the gut physiology by shortening, thickening and atrophy of the villi and increasing the number and size of goblet cells [71,72]. Moreover, changes in the microbial profile induced by high digesta viscosity can cause bile acid deconjugation through microbial activity and impair lipid digestion [73,74], consequently lowering the energy value of barley.

Barley β-glucan consists of d-glucose molecules joined by (1→3) and (1→4) glycosidic bonds and the structure of the glucose chain depends on the relative number of (1→3) and (1→4) β-glycosidic bonds between the repeating glucose units [7]. β-glucan makes up 70% of the endosperm cell wall that surrounds starch granules and about 25% of the aleurone cell walls [49,75].

High β-glucan content is the most detrimental anti-nutritional factor in barley. The content and properties of β-glucan play key roles in determining the potential of barley utilisation by poultry [76]. Higher contents of β-glucan in waxy and high amylose types compared to normal starch, regardless of the absence or presence of hull, have been reported [19,20,28,47]. Comparing two barley types that differed in both hull type and starch type, Perera et al. [13] reported higher total β-glucan in waxy starch hull-less barley compared to normal starch hulled barley (68.6 vs. 38.5 g/kg DM). Izydorczyk et al. [20] compared the total and soluble β-glucan contents in different hull-less barley types and, reported significant differences in total β-glucan, with average values of 75, 69, 63, and 44 g/kg DM for high amylose, waxy, zero amylose waxy, and normal starch barley, respectively. The solubility of β-glucan in high amylose barley was relatively low (21−30%) compared to that in normal (30−44%), zero amylose waxy (34−53%), and waxy (37−53%) barley types [20]. On the other hand, Beames et al. [52] demonstrated that neither the S.NSP nor β-glucan contents differed in hulled and hull-less barley types. The wide range of solubility of β-glucan in different barley types (Table 3) suggest that anti-nutritive properties generated by β-glucan cannot be predicted if only the total content is analysed.

Though the influence of genetic [77] and environmental [78] factors on levels of β-glucan have been established to a great extent, literature on the relationships among β-glucan and other barley constituents have been inconsistent [5,11,20,28]. The wide variability of β-glucan content and solubility, and unpredictable relationship with other components of barley, suggest the importance of assessing the anti-nutritive components of barley prior to feed formulation. The established crucial role of β-glucan in determining the feeding value of barley for broilers [13,34,35] emphasises the need to consider the β-glucan content when selecting barley cultivars for use in poultry diets.

In contrast to β-glucan, arabinoxylans are mainly located in aleurone cell walls, outer layers of barley kernel and husk, and only a small amount is present in endosperm cell walls. The structure of arabinoxylan is composed of two pentosans, namely arabinose and xylan [48]. Holtekjølen et al. [28] observed high contents of arabinoxylan in hulled barley types with a greater insoluble portion (89% of total arabinoxylan), compared to hull-less barley types, and confirmed the presence of arabinoxylans mainly in the hull. Generally, arabinoxylans constitute only a minor portion of water-extractable polysaccharides in barley [49,79] and consequently have received less attention from poultry nutritionists compared to β-glucan. Most of the arabinoxylans in cereal grains are insoluble in water because they are anchored in cell walls by strong cross-links and that arabinoxylans not bound to the cell walls can form highly viscous solutions [48]. Therefore, the influence from arabinoxylans cannot be totally disregarded in the case of barley and measures should be taken to minimise their anti-nutritive effects. 

The molecular characteristics of β-glucan and arabinoxylans play a critical role in determining their physical properties (extractability, viscosity and gelation) and their behaviour in the gastrointestinal tract [16]. Investigating the structure and physicochemical properties of β-glucans and arabinoxylans isolated from hull-less barley, Storsley et al. [33] highlighted that molecular differences of NSP affect their physiological properties and result in different nutritional characteristics, even when the amounts of S.NSP were equal. Perera et al. [80,81] reported that increasing barley inclusions in wheat-based diets reduced the intestinal digesta viscosity despite the higher content of β-glucan in barley compared to wheat and suggested the contribution of factors other than β-glucan concentration on intestinal digesta viscosity of birds fed barley-based diets. It has been shown that digesta viscosity is dependent not only on the concentration of NSP, but also on the molecular weight [82,83]; therefore, a grain with a low content of S.NSP might result in high viscosity if the NSP is of a higher molecular weight [84,85].

#### 3.2.4. Fat and Minerals

Fats or lipids can be considered as the third storage materials in barley grain after starch (513 to 642 g/kg DM [28]) and proteins (108 to 136 g/kg DM [27]) with an average content of 33 g/kg DM [12]. Moreover, barley fats show a little variability according to Svihus and Gullord [4], who reported a narrow range (26−32 g/kg DM) of crude fat for five barley types. Earlier studies on improving the feeding value of barley for poultry birds have emphasised the potential of increasing the intrinsic energy content by increasing storage fat content of barley grains [86,87]. However, no significant increase of fat content has been observed over the years according to Liu [12] and Fedak and Roche [87], who reported fat contents of 33 and 31 g/kg DM, respectively.

Higher content of fat in hull-less barley types compared to hulled types was attributed to the concentration effect caused by the absence of hull [12,13,34,47,88]. Regardless of the hull type, a higher fat content in high amylose barley followed by waxy and normal starch types has been reported [19,88]. Compared to other nutrients, the relationships among fat and other compositional constituents in barley is relatively unexplored, which might be due to the narrow range of fat content resulting in minimal chance of significant effects. 

The major fatty acids (FA) in barley grain are linoleic (518 g/kg of total FA), followed by palmitic (248 g/kg of total FA) and oleic acid (142 g/kg of total FA). The corresponding values in a wheat sample with 22 g/kg DM fat were 597, 203 and 123 g/kg of total FA for linoleic, palmitic and oleic acids, respectively [12]. The high concentration of linoleic acid as an essential FA can be considered as one of nutritional importance in barley grain. In contrast to the relatively constant fat content in different barley types, a wide range of barley FA composition has been reported [87,89] and mainly attributed to oxidation and thus, differences in the storage periods and analytical methodologies. 

Fat in the barley grain is largely concentrated in germ and bran region, while inner endosperm has much less fat [12]. This observation provides scientific basis for the pearling of barley as the removal of surface layers (bran) of grains reduces the lipid content and can improve the storage stability of pearl barley. In addition, Liu [12] proved that removing surface layers improve the stability of FA composition of the remaining kernels by increasing saturated FA while decreasing unsaturated FA.

Most studies providing the mineral composition of different barley types lack information on hull and starch type (Table 4). Rodehutscord et al. [27] and Perera et al. [13] reported potassium as the major mineral followed by phosphorus. Rodehutscord et al. [27] reported a higher content of calcium in barley (ranging from 0.35 to 0.60 g/kg DM) compared to maize (0.04 g/kg DM) and wheat (0.4 g/kg DM). Moreover, barley has a higher sodium content compared to wheat [13] and maize [27]. Except for calcium and sodium, the patterns of differences in other minerals in barley, maize and wheat seemed to be inconsistent. Even though different barley genotypes are commonly analysed for major nutrients and NSP, the potential variation of mineral content in different barley types has been least recognised.

## 4. Barley in Poultry Nutrition

Research into barley use in poultry diets has a long history. According to available literature, around 1930s, studies began to emerge comparing barley with other cereal grains for poultry nutrition [92]. The occurrence of wet litter and sticky droppings was first to be noticed as problems associated with feeding barley-based diets. In addition, depressed growth performance and nutrient utilisation of birds fed barley-based diets were observed [10]. Early research acknowledged a close relationship between extract viscosity of barley and growth impairment of birds and the greater digesta viscosity in birds fed barley-based diets was attributed to the NSP present in barley [76,93]. Enzyme preparations were proven to be effective in ameliorating the depressions in growth and nutrient utilisation in birds fed barley-based diets [94,95]. However, the increased interest of the barley usage in poultry feed due to the development of feed enzymes was challenged by the variable responses of birds fed enzyme supplemented barley-based diets [96]. Moreover, the demand for barley as poultry feed has always been inconsistent, presumably driven by changes in economic circumstances [10,97]. In consequence, the choice of other cereals that are less problematic and maybe more economical has restricted the proportion of barley used in poultry diets to less than 1.0% of total barley utilised as animal feed [8]. In this section of the review, it is aimed to discuss the impact of barley in broiler diets on growth performance, nutrient utilisation and gut morphometric parameters, with emphasis on strengths and weaknesses of previous studies.

### 4.1. Intestinal Digesta Viscosity

Inclusion of barley in poultry diets impedes the nutrient digestion through increasing intestinal viscosity, resulting in inefficient mixing of digesta and enzymes [98]. Transport properties of nutrients at mucosal surface can also be adversely affected, lowering the efficiency of the nutrient absorption [1]. White et al. [93] isolated β-glucan from barley and added it to a maize-based diet and the resultant increase in the intestinal digesta viscosity supported the fact that the β-glucans of barley are the primary cause of poor growth performance. Moreover, it was recognised that not only the concentration but also the structure and molecular weight of NSP is responsible for increased viscosity of the intestinal contents of birds fed barley-based diets [84,85,99]. 

Carré et al. [100] reported that rye resulted in the highest viscosity of gut contents, followed by barley, triticale, wheat, maize, and sorghum. In agreement, majority of the studies has reported more viscous intestinal contents in birds fed barley-based diets compared to birds fed maize-, wheat- and sorghum-based diets (Table 5). However, Shakouri et al. [101] reported higher digesta viscosity in the broilers fed wheat-based diets (5.74 cP) compared to barley-based diets (2.92 cP) speculating that the wheat used to be a viscous cultivar. In addition to the proven influence of barley S.NSP, a variety of factors can influence barley viscosity: (i) grain-related factors such as growing location [102] and storage time [103], (ii) dietary factors such as inclusion level [80,81,104,105], heat processing of grain [106], conditioning temperature during pelleting process [107,108], and (iii) bird-related factors such as the age [109] and sampling point in gastrointestinal tract (GIT; Table 5) [109,110].

Increasing intestinal digesta viscosity of broilers in response to increasing inclusions of barley in a maize-based diet has been reported [104]. Increasing barley inclusions from 300 to 600 g/kg with no enzyme supplementation was shown to increase the digesta viscosity by 222% (from 4.68 to 15.08 cP). However, when a combination of β-glucanase and xylanase was used, the viscosity increased only by 62% (2.44 to 3.95 cP) over a similar increment of barley. Increases in the duodenal digesta viscosity in response to complete replacement of maize with barley, with a greater response in young broilers (d 21) than 42-d old broilers have been reported [105,113]. However, increasing inclusions of barley in a wheat-based diet reduced the intestinal digesta viscosity despite the higher content of β-glucan in barley compared to wheat [80,81]. Perera et al. [80] reported jejunal digesta viscosity reducing from 4.99 to 2.81 cP in response to increasing inclusions of normal starch hulled barley from 0 to 565 g/kg in wheat-based broiler starter diets. Increasing inclusions of waxy starch hull-less barley in a wheat-based broiler starter diet from 0 to 260 g/kg reduced the jejunal digesta viscosity from 4.99 to 3.51 cP [81]. These findings contrast with most previous literature and imply the contribution of factors other than β-glucan concentration to the intestinal digesta viscosity of birds fed barley-based diets.

An increase of intestinal digesta viscosity in response to heat processing of barley grain was reported by Gracia et al. [106], and the reduction of digesta viscosity in response to the added enzyme was greater in heat-processed barley diets. Samarasinghe et al. [107] reported greater dietary viscosity due to high conditioning temperatures (75 and 90 °C) during pelleting a barley-maize-soybean meal diet compared to 60 °C. A supplemental enzyme reduced the dietary viscosity by 11, 14 and 17% in diets conditioned at 60, 75 and 90 °C, respectively, showing a greater magnitude of response at higher temperatures. In agreement, Perera et al. [108] reported that barley-based diets (565 g/kg) conditioned at 88 °C resulted in 10.1% (0.32 cP) higher digesta viscosity compared to the diets conditioned at 60 and 74 °C, and impaired nutrient utilisation and bird performance. The supplemental carbohydrase, however, did not reduce the intestinal digesta viscosity. The variable response of digesta viscosity to supplemental carbohydrase in different studies emphasises the need for the correct determination of enzyme dosage in barley-based diets, with close attention to feed processing conditions to increase the barley inclusion in poultry diets by strategically minimising the viscosity related negative consequences.

Decreasing intestinal viscosity with increasing age of the broilers fed barley-based diets has been reported in some studies [106,109,114]. Intestinal viscosity is not a major limiting factor in adult birds as it is in young birds fed barley-based diets [98], as older birds have sufficiently developed GIT to overcome the negative effects of high β-glucan-induced digesta viscosity [95]. Evaluating a high viscosity hull-less barley, Salih et al. [114] reported that the relative intestinal digesta viscosity dropped from 2.59 at 14 d to 1.74 in 56 d broilers. Petersen et al. [109] reported that foregut digesta viscosity of broilers fed barley-based diets reduced with age by 51%, from 16.7 cP at 25 d to 8.2 cP at 45 d. These observations support the suggestion by Bedford [115] that the mechanisms of viscosity needed to be re-evaluated as being a function not only of the cereal being fed, but also of the age of the bird that would be more relevant of NSP rich barley-based diets.

### 4.2. Growth Performance

The growth performance in broilers fed barley-based diets has been reported to be poorer compared to maize [116,117], wheat [114,118] and sorghum [119], and commonly attributed to the greater digesta viscosity in barley-fed birds. Shakouri et al. [101] and Tang et al. [119] evaluated barley as the sole cereal in the broiler diets in comparison to maize, sorghum and wheat and reported that birds fed barley-based diets had the lowest weight gain (WG), feed intake (FI) and poorest feed to gain ratio (F/G). In contrast, Brenes et al. [120], who compared barley (cultivar, Scout) with wheat in broilers, reported 58 g superior WG for barley-fed birds at 42 d. However, the F/G of birds fed barley-based diets was impaired by 8 points. The WG and F/G differences caused by the grain type were minimised by the supplemental carbohydrases.

Due to the low metabolisable energy content of barley (Table 6), birds need to consume more feed to maintain a constant energy intake [121]. However, reduced feed passage associated with higher digesta viscosity caused by NSP [114] can depress the FI, especially in younger birds [122], resulting in birds not being able to meet their nutritional and energy requirements [6]. Moreover, barley is less palatable to poultry compared to maize [123] and wheat [124]. The removal of the hull is believed to increase the palatability of barley [113] and this perception was one of incentives for the development of hull-less barley types.

The effect of barley cultivar on the growth performance of birds has been previously evaluated. Bergh et al. [35] compared hulled barley cultivars (696 g/kg) with normal, high amylose and waxy starch types, without or with supplementation of β-glucanase, for young broilers. Birds offered normal starch barley had a better BW, FI and F/G. The magnitude of improvement in growth performance due to the supplemental enzyme was greater in birds fed high amylose and waxy barleys. The increases of WG in 13-d old birds in response to supplemental enzyme were 22, 44, and 38 g/bird for normal, high amylose and waxy barley types, respectively, and the corresponding improvements in F/G were 7, 24 and 21 points, respectively. 

Yu et al. [113] evaluated the inclusion of de-hulled barley at inclusion levels of 0, 400 and 800 g/kg, and supplementation of β-glucanase in iso-nitrogenous and iso-caloric maize-based diets and reported increased FI and WG with no effect on feed efficiency in response to the increasing inclusion of barley. The effect of barley inclusion in poultry diets on feed efficiency has been inconsistent, as both improvements [118] and depressions [128] have been reported. Moss et al. [128], replaced wheat (w/w basis) with 0, 272, 408 and 544 g/kg of waxy starch hulled barley in broiler diets with no enzyme addition and reported that increasing levels of barley consistently decreased the WG, but had no effect on F/G. Classen et al. [129] substituted hull-less barley (starch type not identified) on weight basis (0, 200, 400 and 600 g/kg) for wheat in a broiler starter diet and reported a linear decrease in BW with increasing levels of barley, while no changes were reported for F/G.

Friesen et al. [118] studied different inclusions of hull-less barley (0, 350 and 700 g/kg) in a wheat diet. Weight gain and F/G of 14-d old broilers fed the hull-less barley at 350 g/kg was similar to those fed the control wheat diet, wherein barley inclusion at 700 g/kg resulted in the lowest WG and highest F/G. The deterioration of growth performance associated with barley inclusion reported in previous studies may partly be explained by weight-to-weight substitution of barley for the major cereal in the diets [118,128], resulting in lower metabolisable energy and digestible AA content than the corresponding cereal-based diets. 

With the recognition of the importance of using nutrient profiles specific to each barley cultivar, Perera et al. [13] replaced wheat with different levels of normal-starch hulled barley using apparent metabolisable energy (AME) and standardised digestible AA contents specific to the tested barley. These researchers reported that the WG of birds increased up to 283 g/kg of normal-starch hulled barley inclusion in a wheat-based diet and then decreased with further barley inclusions. The F/G, however, was improved with increasing barley inclusions in diets. In a follow up study, Perera et al. [81] evaluated the increasing inclusions of a waxy starch hull-less barley in a wheat-based diet, and reported that waxy starch hull-less barley could replace wheat up to 260 g/kg with no negative effect on growth performance and feed efficiency. Careful manipulation of optimum feed processing conditions and enzyme supplementation may further minimise the inherent variability of barley grains and hence allow increased inclusion of barley in broiler diets. The optimum barley inclusion level in broiler finisher diets remains comparatively unexplored. With the ability of mature birds to counteract the adverse effects of barley-based diets, the potential of higher barley inclusion levels in broiler finisher diets can be speculated.

### 4.3. Energy Utilisation

Metabolisable energy of a cereal grain is dependent on the energy contained, the availability of the energy to the bird, and the presence or absence of anti-nutritive factors such as S.NSP [130]. Wide variations in the AME within and between grain types is primarily attributed to their variable chemical and physical characteristics [9] and grain specific anti-nutritive factors [21]. Kocher et al. [131] reported the AME of Australian barley types to range from 10.4 to 12.2 MJ/kg DM. In addition, Choct et al. [126], who analysed 11 barley cultivars, reported ranges of 11.6 to 13.8 and 12.5 to 13.58 MJ/kg DM for AME of barley in broilers and layers, respectively. Among the cereal grains, barley has been identified as one of the most variable cereal grains in terms of its energy value [126] and this variability is not reflected in feed composition tables [1,130].

Early studies evaluating the feeding value of barley for poultry attributed its lower energy content to the presence of fibrous hulls [10]. However, Scott et al. [130] analysed 14 barley types characterised for hull type, starch type, malting and row (two- or six-row) and reported the lack of effect from hull type on AME in non-enzyme supplemented barley diets. It was speculated that the adverse effects of the higher fibre content of hulled cultivars on AME were confounded by the higher β-glucan levels of the hull-less cultivars. In carbohydrase supplemented diets, however, hull-less barley cultivars showed greater AME content due to the carbohydrase enzyme action on minimising anti-nutritive effects of β-glucan.

As shown in Table 6, comparing two hull-less barley types that differed in starch type, Ravindran et al. [34] reported 1.74 MJ/kg higher AMEn content for the normal starch barley than waxy starch barley. On the other hand, comparing two normal starch barley types differing in the presence of hull, only 0.25 MJ/kg difference in AMEn was reported. This finding suggests that starch characteristics of barley cultivars are probably more important than fibre contents in determining the available energy content of barley for broilers. In contrast, Villamide et al. [9] compared the energy content of eight barley cultivars, without and with a multi-component enzyme complex, and reported no relationship between AMEn of non-supplemented barley cultivars and chemical composition. Perera et al. [13] reported a greater AMEn value for normal starch hulled barley than waxy starch hull-less barley (13.39 vs. 10.60 MJ/kg DM); it was difficult to speculate the most critical factor causing the AMEn difference, as both starch and hull contents differed in tested barley types.

The available energy of cereal grains has a strong negative correlation with NSP concentrations in each grain type [125]. In the case of barley, especially in non-enzyme supplemented diets, the bioavailable energy depends on its content of soluble β-glucan and consequent higher digesta viscosity [95]. A linear reduction of AMEn with increasing inclusions of barley in wheat—[118] and maize—[104] based diets was reported and attributed to the increasing digesta viscosity. Villamide et al. [9] demonstrated a 0.14 MJ decline in dietary AMEn for each 100 g/kg increase in barley inclusion. In contrast, linear increases in dietary AMEn in response to increasing inclusions of normal starch hulled barley and waxy starch hull-less barley in wheat-based diets have been reported [80,81]. The reported improvements in energy utilisation were mainly attributed to the decreasing jejunal digesta viscosity in response to increasing inclusions of barley. Fuente et al. [104] reported a 0.06 MJ decline in AMEn per unit (cP) increase in digesta viscosity, suggesting that digesta viscosity accounts for 97% of the variation in AMEn among barley-based diets and supported the recent findings [80,81].

### 4.4. Nutrient Digestibility

#### 4.4.1. Protein and Amino acids

Owing to the high inclusion of cereal grains in poultry diets, cereals contribute up to 35% of the total dietary CP and have a substantial contribution to the supply of dietary AA. To increase the dietary inclusion of barley without any adverse effect on performance, the factors affecting AA digestibility in birds fed barley-based diets should be well-understood. The digestibility of AA has been determined either as apparent or standardised [132]. Several studies have evaluated the apparent ileal digestibility (AID) of AA in different barley types [34,133,134,135]. However, studies evaluating the standardised ileal digestibility (SID) of AA in different barley types are limited [13,46,136,137,138]. As shown in Table 7, the SID AA of maize, triticale, sorghum and wheat are higher than that of barley. Barua et al. [138] reported average SID AA for maize, sorghum, wheat and barley as 0.838, 0.804, 0.778 and 0.723, respectively. The AA digestibility of barley ranged from 0.639 for lysine to 0.815 for cysteine [138].

The incomplete digestion of AA justifies the use of digestible AA values, instead of total AA values, for broiler feed formulations. The AID AA, however, is not recommended to be used in diet formulations due mainly to the underestimation of AA digestibility caused by endogenous AA flow and less additivity, compared to SID AA, in complete diets. These concerns are more critical for low-protein feed ingredients [139,140] such as barley, and therefore, the use of SID AA values with higher precision and additivity is recommended for barley-based diet formulations. 

Inconsistent AA digestibility in different barley cultivars has been reported [34], and partly attributed to differences in the concentrations of NSP. The average AID values reported for non-supplemented hulled normal starch, hull-less normal starch, and hull-less waxy barley-1 and hull-less waxy barley-2 were 0.67, 0.66, 0.63 and 0.71, respectively, with corresponding CP contents of 116, 104, 105 and 137 g/kg DM, respectively [34]. According to Perera et al. [13], who compared two barley types that differ in CP content, average SID AA in normal starch hulled barley were superior to waxy starch hull-less barley (0.787 vs. 0.740), despite the higher concentrations of AA in the latter (101 vs. 133 g/kg DM). The inter-cultivar variability of AA digestibility justifies the use of SID values, specific to each barley type, for formulating balanced barley-based diets, ensuring an adequate supply of AA for maintenance and growth functions.

Significant improvements in AA digestibility of barley due to exogenous carbohydrases have been reported [34,84,141]. However, the effect of enzyme supplementation on individual AA has been inconsistent. Other factors that contribute to the variation of protein and AA digestibility in barley-based diets include; bird type [135], age of birds [137], barley particle size and feed form [138] and thus warrants consideration of these factors when determining AA digestibility in barley grain.

#### 4.4.2. Starch 

Supported by the similar trends between starch digestibility and energy utilisation [13,34,80,81,142], digestible starch is considered as the primary contributor to metabolisable energy in barley-based diets. Table 8 shows the comparison of ileal starch digestibility in broilers among different grain types and barley types. While starch in maize is almost completely digested in broiler chickens [143], other cereal grains show comparatively lower starch digestibility and greater variability than maize. Reasons for this variability include variations in starch granule structure, anti-nutritional factors and access problems in coarse particles and are extensively reviewed [31,100,143].

Barley grains can be categorised based on the starch type and, in contrary to the expectation that waxy barley starch with more amylopectin (970−1000 g/kg of starch [29]) is more digestible [37], poor starch digestibility has been observed in birds fed waxy barley-based diets, regardless of the hull type [13,34,35]. This finding is suggestive of the contribution of factors other than starch type and hull type, in particular β-glucan content, affecting starch digestibility in barley.

Ankrah et al. [110] evaluated the starch digestibility in birds fed hull-less barley cultivars of normal or waxy starch (722 and 945 g/kg amylopectin, respectively) and, despite the higher digesta viscosity of birds fed waxy starch barley compared to the normal starch barley (276 vs. 102 cP), reported similar starch digestibility for the different barley cultivars. Poor response of starch digestibility to variations in digesta viscosity in other grains has been previously reported [145] and among the three main nutrients (N, starch and fat), the extent of digestibility reduction due to viscosity seems to be lowest for starch [146,147,148]. However, Carré [100] suggested that viscosity may induce a noticeable effect on starch digestibility in high viscosity barley types. Nevertheless, owing to the lack of sensitivity of starch digestibility to digesta viscosity [149], it was hypothesised that the effect of exogenous enzymes on starch digestion cannot be explained only on the basis of reduction of intestinal digesta viscosity [100]. With the finding by Andriotis et al. [15] that endosperm cell wall degradation is an important determinant of starch degradation rate in barley grains, it can be speculated that the supplemental carbohydrases enhance the starch digestibility primarily by breaking down the barley endosperm cell walls and releasing the encapsulated starch granules. 

Enhanced starch digestibility of barley-based diets in response to supplemental carbohydrases has been commonly observed in studies with broilers [13,34,35,80,81,108]. According to Ravindran et al. [34], the magnitude of improvement in starch digestibility varied depending on barley type and was markedly greater in waxy genotypes (41and 73%) compared to the normal starch genotypes (15 and 18%). In agreement, Perera et al. [13] reported a greater starch digestibility response to enzyme supplementation in waxy starch hull-less barley than normal starch hulled barley (7.4 vs. 0.51%), which was attributed to differences in β-glucan content (68.6 vs. 38.5 g/kg DM).

Feed processing techniques can have variable outcomes on starch digestibility depending on the grain type [143]. Ankrah et al. [110] reported enhanced starch digestibility in reground pellets compared to mash (0.860 vs. 0.774) in broilers fed barley-based diets, irrespective of the starch type and enzyme supplementation. Enhanced starch digestibility in response to replacing ground barley with whole barley (WB) was reported [150]. However, Perera et al. [149] reported that starch digestibility in broilers fed barley-based diets was not influenced by barley particle size. In agreement, Tari et al. [151] reported no influence of the method of barley inclusion (fine, coarse and WB) in wheat-based diets on starch utilisation. Enhanced starch digestibility may also be attributed to mechanisms facilitated by well-developed gizzards. However, no impact on the starch digestibility was reported despite functional gizzards in birds fed coarse [149] and WB [151]. On the other hand, a recent study [152] reported an enhanced starch digestibility in response to increasing the WB inclusions from 0 to 141 and 282 by 3.6 and 5.7%, respectively, despite the lack of impact on gizzard size. In addition, starch digestibility in broilers fed pelleted barley-based diets conditioned at 88 °C was impaired compared to those fed diets conditioned at 60 °C [108]. 

#### 4.4.3. Fat

Increased intestinal digesta viscosity in birds fed barley-based diets has been reported to be more detrimental to fat digestion [98,153], as fat reported to be the nutrient most affected by the presence of S.NSP in the diet [146]. High digesta viscosity limits diffusion and passage of droplets of emulsion, fatty acids, mixed micelles, bile salts and lipase within the gastrointestinal tract, leading to reduced transport of micelles to the mucosal surface [154,155]. Martinez et al. [156] suggested that in addition to S.NSP, fat-soluble tocotrienol (subclass of vitamin E) present in barley can inhibit the cholesterol synthesis exacerbating the bile acid shortage created by S.NSP. In addition to the adverse impact by higher intestinal digesta viscosity, stimulation of gut microbial growth [71,114] that leads to higher bacterial activity may reduce the recycling of bile acids and the resultant low concentration of bile salts in birds fed barley-based diets, leading to poor digestibility of fat.

Bergh et al. [35] who compared three hulled barley types differed in starch type, reported no differences in ileal fat digestibility between barley types, despite the different amounts of S.NSP. However, supplementation of β-glucanase enhanced the digestibility of fat with the greatest magnitude of response observed for waxy barley types. Friesen et al. [118] evaluated the impact of increasing inclusions of hulled and hull-less barley cultivars in a wheat-based diet (on a *w*/*w* basis and similar fat inclusion) and reported decreasing fat digestibility only in broilers fed hull-less barley. The depressed fat digestibility was, however, restored with supplemental carbohydrases. 

Viveros et al. [71] reported a lower fat digestibility in 12-d old broilers compared to 28-d old broilers (73.2 vs. 83.2%), suggesting an age-related response in fat digestibility in broilers fed barley-based diets. Limited production of lipase [157] and bile salts [71] causing lower fat digestibility has been reported in very young broilers fed barley-based diets. Supplementation of barley-based diet with β-glucanase elevated the lipase activity in both young broilers and adult roosters with a greater magnitude in the former [98].

The digestibility responses to barley inclusion in broiler diets seems to be nutrient-dependent due to variable sensitivity of nutrients to digesta viscosity, the storage location of nutrients and interactions with other nutrients. Overall, a small change in the concentration and/or the molecular weight of S.NSP in barley can significantly impact nutrient utilisation. Therefore, the prediction of digestibility response from the bird’s capacity to utilise the nutrients solely from the nutrient composition data is challenging and justifies the need for using digestible nutrient values, especially AA, in barley-based diet formulation. Moreover, determining the rate and extent of nutrient digestion in different barley types will enable the strategic manipulation of diet formulation, feed processing and enzyme supplementation and, consequently, increasing the inclusion of barley in poultry diets.

### 4.5. Intestinal Morphology

Digesta viscosity can cause significant influence on the intestinal morphometry of birds fed barley-based diets. Viveros et al. [71] reported shortening, thickening and atrophy of villi, and increased number (hypertrophy) and size (hyperplasia) of goblet cells in the jejunum of birds fed barley-based diets (600 g/kg) compared to those fed maize-based diets. These effects were minimised, however, by supplementation with β-glucanases. Onderci et al. [117] also reported shorter and narrower villus in birds fed barley- compared to maize-based diets. Shakouri et al. [101] reported decreased jejunal villus height and villus: crypt ratio in birds fed diets with 600 g barley/kg compared to the diets containing maize, wheat and sorghum (623 g grain/kg). Kalantar et al. [158] observed shorter villus height in birds fed diets with barley included as low as 150 g/kg in a maize-based diet. The poor growth performance of broilers fed barley compared to other grain types was attributed to alterations of intestinal morphology induced by barley inclusion [71,159,160]. Comparative studies based on different barley cultivars on intestinal morphometry are limited. Barley inclusion in broiler starter diets increased the jejunal villus height supporting the nutrient utilisation [80,81], that can be attributed to the decreasing digesta viscosity in response to increasing barley inclusions in the diet. In parallel, Karunaratne et al. [161] reported that the hull-less barley decreased the villi height compared to wheat in 33-d old broilers, which was attributed to the damage on epithelial villi in the ileum by higher digesta viscosity.

Comparing barley with wheat for relative lengths and weights of the GIT segments, Brenes et al. [120] reported longer duodenum, jejunum, ileum and caeca and lighter gizzard in birds fed barley-based diets than those fed wheat-based diets. While supplemental enzymes did not impact the gut morphometry of birds fed wheat-based diets, it reduced the lengths of intestinal segments in barley-fed birds. Comparing two hull-less and hulled barley cultivars, heavier proventriculus and gizzard and, shorter jejunum and ileum were reported in birds fed hulled barley than those fed hull-less cultivar [120]. 

A greater gizzard weight in response to increasing barley inclusion in a wheat-based diet [80,81] was attributed to higher I.NSP content in the diets with greater barley inclusions [80]. However, when the gizzard weight increased with increasing waxy starch hull-less barley in wheat-based diets in Perera et al. [81], dietary I.NSP content did not seem to be influential. This led to the speculation that the higher level of β-glucan in waxy starch hull-less barley (68.6 g/kg [13]) would have positively contributed to barley grain hardness [162] and subsequently to gizzard development. This speculation was supported by the microscopic images with thicker endosperm cell walls for waxy starch hull-less barley [13]. 

The particle size of the barley can be manipulated to benefit the gizzard development of birds. A 19% increase in the relative weight of gizzard was reported in response to the increasing barley particle size from fine to coarse (2.0 vs. 8.00 mm [149]). Tari et al. [151] compared different methods (fine, coarse and WB) of barley inclusion (283 g/kg) in a wheat-based diet and reported reduced weights of crop, proventriculus, jejunum and ileum, but greater gizzard weights in WB than fine and coarse-barley diets. In contrast, a recent study in our laboratory [152] reported no impact of WB inclusion at 141 and 282 g/kg in barley-based diets on gizzard development. 

### 4.6. Welfare and Health

Incorporation of viscous cereals such as rye, barley, triticale and wheat in poultry diets has been associated with litter problems caused by elevated excreta moisture and increased occurrence of sticky droppings. Roberts et al. [163] compared the effect of sorghum, barley, wheat and triticale on excreta moisture content in laying hens and reported that barley diets resulted in the wettest litter (77.5 vs. 74.5% moisture), a finding primarily attributed to increased digesta viscosity that lowers water absorption, increasing the water loss through the excreta. This situation has led to welfare and management problems in barley-fed birds. Dirty eggs in layers and breast muscle damage in broilers resulting from sticky droppings reduce the marketability of eggs and chicken [6,70,96,164]. The occurrence of foot pad dermatitis (FPD) characterised by necrotic lesions on the plantar surface of feet in growing broilers and turkeys is promoted by wet litter and is considered as a major welfare issue in birds fed barley-based diets. Moreover, increasing litter moisture caused by the sticky droppings can reduce the air quality of the poultry house [1]. Increasing dietary inclusion of barley increased the water consumption and the incidence of sticky droppings but these effects were diminished with the supplemental enzymes [165].

The FPD can impair the health and productivity of birds and reduce the quality of chicken feet as human food resulting in economic losses [166,167]. Litter moisture less than 30% is usually recommended as optimal for footpad health [166]. High risk of FPD in broilers fed barley-based diets is anticipated due to the occurrence of sticky droppings and the continuous sticking of excreta deteriorating the epidermis and keratin layers in the footpad [168]. Cengiz et al. [167] evaluated barley inclusion at 250 g/kg in a maize-based diet, without and with enzyme supplementation, on the FPD in broiler chickens exposed to early high-moisture litter from d 1 to 5 and reported no influence of barley inclusion on development of FPD, litter moisture level, or litter pH. In a follow-up study, Cengiz et al. [169] included hulled barley at 300 g/kg in a maize-based diet and observed high litter moisture (32 vs. 19%) and high incidence and severity of FPD in barley-fed birds in comparison to the birds fed maize-based diet at 42 d of age. The occurrence of FPD, however, cannot be solely attributed to the inclusion of NSP rich ingredients in the diet and, seemed to be influenced by litter properties and management conditions as well. Predisposing factors created by the dietary inclusion of barley can be managed through proper management practices and dietary modifications. However, literature on the efficacy of nutritional approaches on the litter quality and FPD incidence are inconsistent.

Barley β-glucans can modify the intestinal microflora composition leading to increased susceptibility to diseases [7]. Chickens fed barley-based diets have been reported with an increased incidence of necrotic enteritis associated with increased levels of *Clostridium perfringens* [170,171]. It is reasonable to assume that a slower passage rate caused by high intestinal viscosity can facilitate the colonisation of potentially pathogenic bacteria [172], deteriorating the health of barley-fed birds. 

### 4.7. Age of Birds 

Bird age is a factor determining for feeding value of barley because of the influence on intestinal digesta viscosity. The reduction in the effects of digesta viscosity with advancing age [109] suggests that the impact of barley NSP is age-dependent due to changes in the GIT. According to Almirall et al. [98], the production and functionality of digestive enzymes in young chicks are disturbed by viscosity. However, when diets were supplemented with enzymes, young birds had a greater response to β-glucanase [98]. It has been suggested that mature birds have a sufficiently developed GIT to counteract the negative effects of the β-glucans [95,114,122]. 

Salih et al. [114] reported that WG and feed efficiency of broilers fed wheat, hull-less barley and enzyme supplemented hull-less barley diets were not influenced after four weeks of age. Viveros et al. [71] also reported lower fat and starch digestibility in 12 d-old-broilers compared to 28 d-old broilers fed barley-based diets. These observations highlight the importance of considering bird age when determining the optimum barley inclusion and enzyme dosages in broiler diets as an important factor that can influence nutrient digestibility and performance responses.

### 4.8. Recommended Inclusion of Barley in Broiler Diets

A wide range of inclusion levels of barley has been recommended for broiler diets. However, recommendations on the optimum inclusion of barley have been contradictory due to confounding factors such as starch type, presence of hull and cultivar differences, which have been overlooked in most previous studies. As shown in Table 9, most studies have replaced other cereals with barley either on a weight-to-weight basis [105,128,173,174] or by using nutrient composition data for barley and the substituted grain from established data sources such as NRC [116] and tables published by Spanish Foundation for the Development of Animal Nutrition (FEDNA) [175,176], or chemical analysis [177]. Studies where barley-based diets were formulated using accurate nutrient profiles specific to the barley cultivar based on determined AME and digestible AA contents are scanty. 

According to previous studies, Arscott et al. [173] suggest that barley can be included in broiler diets up to 153 g/kg without affecting growth performance. Jeroch and Dänicke [10] recommended up to 200−300 g barley/kg for broiler finishers. Brake et al. [177] suggested that 200 g barley/kg can be included in both broiler grower and finisher diets without compromising growth, feed efficiency or litter conditions. According to Bergh et al. [35] and Yu et al. [105], 140 g barley/kg can be included in β-glucanase supplemented broiler diets. This discrepancy of recommendations for barley inclusion in broiler diets can be partly attributed to the lack of characterisation of tested barley types and inconsistency of research methodology, as shown in Table 9. 

The nutritive value of grains for poultry is determined not only by the chemical and physical properties of grains but also by the interactions of ingestion, digestion, absorption, and metabolism in birds [21]. As discussed in this review, minor changes in NSP content and composition can substantially impact the performance and nutrient utilisation causing considerable variation between barley types. In order to minimise the impact of barley variation and meet birds’ nutrient requirements based on their nutrient utilisation capacity, the use of grain-specific AME and digestible nutrients, particularly AA, when formulating barley-based diets, is strongly recommended. Perera et al. [13] evaluated the optimum inclusion level of normal starch hulled barley in a wheat-based diet using AMEn and standardised digestible AA contents specific to the test barley and reported that WG increased up to 283 g/kg of barley inclusion and then decreased with further inclusion. In Perera et al. [81], the optimum inclusion of waxy starch hull-less barley was evaluated using grain specific AMEn and SID AA values and, maximum waxy starch hull-less barley inclusion (260 g/kg) had no adverse effects on the WG and even improved feed efficiency. The findings by Perera et al. [80,81] lead to recommendations that normal starch hulled barley and waxy starch hull-less barley can be safely included up to 283 and 260 g/kg, respectively, in a balanced, pelleted wheat-based broiler diet, and showed the potential to increase the inclusion of barley in poultry diets when the diets are formulated using the grain-specific metabolisable energy and digestible nutrients, and pelleted.

## 5. Measures to Overcome the Limitations of Barley in Poultry Diets

With growing knowledge of physical and chemical characteristics of barley grain and mechanisms of anti-nutritive action, measures to minimise or even eliminate the anti-nutritive impact of barley NSP in poultry diets have evolved over the years. These measures can be categorised as (i) morphological and compositional changes in barley grains via genetic selection and breeding, (ii) amelioration of NSP-induced anti-nutritive conditions by feed additives and (iii) physical manipulation of barley grains using appropriate feed processing techniques. This section intends to provide a comprehensive review of each measure highlighting the specific objectives, mechanisms, and outcomes.

### 5.1. Genetic Development 

#### 5.1.1. Hull-Less Barley

The established perception around the 1970s that the fibrous hull of barley had a significant anti-nutritive influence on energy utilisation in poultry feeding [17] led to the development of hull-less barley to promote the acceptance of barley as a poultry feed ingredient [1,11]. Use of hull-less over hulled barley in poultry feed also eliminates the cost and labour associated with dehulling, resulting in a cereal that is more compatible with nutrient-dense feeds preferred by the poultry industry [178]. 

Both hulled and hull-less barley types have been reported with variable amounts of nutrients suggesting an inconsistent effect of the hull type on nutrient content. Nevertheless, constant lower concentrations of I.NSP in hull-less barley compared to hulled barley have been reported in different studies [19,34], which eventually equalised hull-less barley to wheat, in terms of fibre content [179].

As shown in Table 3, different β-glucan contents have been reported for hull-less varieties indicating the influence of factors other than presence or absence of hull. Moreover, majority of these studies have neglected the other physico-chemical differences, such as starch type, associated with different hull-less barley cultivars. Ravindran et al. [34] emphasised the need for considering the characteristics of starch and β-glucan content over the fibre content, when selecting barley cultivars for poultry diets.

With the recent recognition of importance of low levels of fibre in poultry diets to restore the gut integrity of birds fed highly processed pelleted diets, the tendency is to incorporate insoluble functional fibres, such as hulls, into poultry diets. The impact of barley hulls on gizzard development has been discussed in the literature [64,180]. Instead of adding the hulls separately, direct use of hulled barley in poultry diets can be considered as a cost-effective approach. 

#### 5.1.2. Waxy-Starch and High Amylose-Starch Barley

In addition to the conventional barley composed of normal starch (650−840 g/kg amylopectin), both hulled and hull-less barley have been developed into waxy (850−1000 g/kg amylopectin) and high amylose (450 g/kg amylose; 550 g/kg amylopectin) barley types [29,30]. These cultivars vary not only in starch composition but also in the morphology and physico-chemical characteristics of starch granule, as discussed in Section 3.2.1. From a poultry nutrition perspective, development of waxy starch barley was considered advantageous in terms of starch digestion. According to in vitro enzyme hydrolysis of barley starches, waxy barley starch has a higher susceptibility to α-amylase, compared to normal or high amylose barley starches [37,38]. However, when analysed in vivo, waxy barley-based diets were found to have a lower starch digestibility (Table 8, [34,35]). In addition, as discussed in Section 3.2.1., birds fed waxy starch barley diets had a poor growth performance compared to those fed other barley types [35]. The impaired growth performance and nutrient utilisation in birds fed waxy starch barley has been attributed to soluble β-glucan with high molecular weights, which occur in greater amounts in waxy starch barley types [33].

Nevertheless, waxy starch barley might benefit feed manufacture in pellet form due to lower starch gelatinisation temperature, resulting in higher physical pellet quality and reduced energy input in pellet production. According to Ankrah et al. [110], equivalent pellet hardness in waxy starch hull-less barley was achieved at a lower temperature (by 14.2 °C) than in normal starch hull-less barley. However, waxy starch barley, with higher soluble β-glucan content, also increased digesta viscosity compared to the normal starch barley. With a comparatively greater efficacy in waxy starch barley types (Table 10), exogenous enzymes are proven to mitigate the anti-nutritive effects of S.NSP, making waxy starch barley an attractive feed ingredient for poultry.

### 5.2. Feed Enzymes

With the developing knowledge on the anti-nutritive impact of barley NSP in poultry diets, research on the use of feed enzymes in barley-based diets has evolved over the years. Initially, supplementation of amylolytic enzymes to barley-based broiler diets was reported to be effective in reducing the sticky droppings and enhancing the growth performance [181,182,183]. At that time, only rudimentary knowledge was available on substrate specificity of exogenous enzymes. However, with the finding by Burnett [76] that viscous β-glucans present in barley are the main reason for its low nutritive value, the observed performance improvement in birds fed barley-based diets by amylolytic enzymes was attributed to a contaminant side activity of β-glucanase and its action of reducing digesta viscosity [184].

Following this recognition [164,185], the first β-glucanase was commercialised in 1984 [186]. When supplementing barley-based diets with exogenous enzyme, the rule of thumb adopted by the poultry industry was “barley + β-glucanase = wheat” [187]. Currently, almost all barley-based broiler diets worldwide are supplemented with glycanases (xylanases and β-glucanases) [188]. Three major modes of action of NSP-degrading enzymes have been recognised in the literature; (i) reduction of digesta viscosity via partial depolymerisation of NSP [98], (ii) release of encapsulated nutrients via cell wall degradation [94,189] and, (iii) improvements in gut microbiota through the generation of prebiotic oligosaccharides [115,190]. However, the improvement in growth performance and nutrient utilisation in response to the supplementation of carbohydrases in barley-based diets has been commonly attributed to the viscosity reduction caused by the partial degradation of S.NSP [98,110]. Moreover, NSP-degrading enzymes can disrupt endosperm cell walls, enabling greater access of digestive enzymes to encapsulated protein and starch [94,115,189]. Supporting the hypothesis of cell wall solubilising effects of added carbohydrase, Ravn et al. [191] has shown the in vitro destruction of the cell walls taking place in barley by supplemental xylanase. 

Exogenous carbohydrases depolymerise high molecular weight β-glucan in barley in a dose-dependent manner and generate fermentable oligosaccharides that can act as prebiotic compounds in the GIT of chickens [161]. Prebiotic oligosaccharides can encourage proliferation of beneficial bacteria such as *Lactobacillius* and Bifidobacteria [74,192] preventing the growth of pathogenic bacteria such as Escherichia coli and Salmonella through competitive exclusion [160,193]. A substantial increase in Bifidobacteria counts in the caecal digesta (from 3.92 to 9.69 log cfu/mL of digesta) [74] and 61% increase in lactic acid production in the crop [73] of broilers fed barley-based diet in response to β-glucanase supplementation has been reported. The improvement in nutrient utilisation due to supplemental carbohydrases in a wheat-barley-based diet has been partly attributed to the reduction of total anaerobic bacteria [160]. 

As shown in Table 10, majority of studies with barley-based diets have confirmed the efficacy of dietary carbohydrase supplementation in enhancing the feeding value of barley for broilers through improved growth performance, enhanced nutrient utilisation and flock uniformity. In addition, supplemental carbohydrases minimise the variability in nutritional value of barley grains. Villamide et al. [9] reported that supplementing a multi-enzyme containing β-glucanase, xylanase, and protease, reduced the range of AMEn variability in eight barley cultivars by 23.9%, with a greater effect on highly viscous barley types. Kocher et al. [131] reported that variability of AME of 11 different barley cultivars was reduced by 55% due to supplemental β-glucanase. 

Combinations of different exogenous enzymes have also been evaluated in barley-based diets (Table 10). Microbial phytase has been used in combination with carbohydrases in barley-based diets [142,149,194]. In addition to primary objectives of adding phytase to facilitate the release of phytate-bound P and to reduce the P effluents from intensive animal production [195], the supplementation of phytase to barley-based diets is justified by the fact that phytate is an integral part of barley cell wall matrix [196]. The combination of enzymes in barley-based diets is believed to facilitate each other’s substrate access. Nevertheless, when a combination of different enzymes is used, the response of barley to enzyme mixtures is largely dependent on content of carbohydrase, especially β-glucanase, over other enzymes [197]. The variable response to supplemental enzymes in birds fed barley-based diets (Table 10) can be attributed to the variable stability of enzymes during feed processing [198], variations in barley anti-nutritional composition, mainly β-glucan [13], starch structure [34,80,81], and interactions with grain physical characteristics (e.g., particle size and hardness) [61,149].

**Table 10 animals-12-02525-t010:** Response of growth performance, nutrient utilisation and intestinal digesta viscosity of broilers fed barley-based diets to supplemental enzymes.

Reference	Barley		Inclusion Level (g/kg of Diet)	Feed Form (M/P) ^3^	Components in Carbohydrase (BG/XY) ^4^	Phytase	Bird Age (d)	Growth Performance ^5^			Nutrient Utilisation ^6^						Reduction in Digesta Viscosity (cP)
	Hull Type ^1^	Starch Type ^2^						WG (%)	FI (%)	F/G (Points)	N (%)	Starch (%)	Fat (%)	P (%)	AME (%)	AME n (%)	
[98]	-	-	600	M	BG	-	24	8.8	3.2	4	12.6	6.91	5.35	-	-	-	11
	-	-						13.2	6.0	3	16.6	2.01	3.14	-	-	-	26
[35]	H	N	696	M	BG + XY	-	13/18 ^7^	8.0	4.6	7	8.2	7.7	22.1	-	-	-	-
		HA						18.6	6.3	24	6.8	7.9	14.1	-	-	-	-
		W						17.6	9.3	21	6.9	12.6	23.4	-	-	-	-
[110]	HL	N	610	M	BG	-	21	54.6	18.3	50	-	52.4	-	-	-	-	245
		N		P				37.6	5.6	50	-	29.6	-	-	-	-	91
		W		M				44.0	7.4	56	-	87.3	-	-	-	-	466
		W		P				51.5	19.4	45	-	21.3	-	-	-	-	267
[34]	H	N	963	M	BG	-	28	-	-	-	17.3	17.9	-	-	-	9.2	-
	HL	N						-	-	-	20.7	15.2	-	-	-	5.5	-
		W						-	-	-	16.5	41.0	-	-	-	22.2	-
		W						-	-	-	14.7	73.0	-	-	-	23.1	-
[194]	-	-	820	M	BG + XY	-	42	-	-	-	-	-	-	-	0.5	-	-
					-	+		-	-	-	-	-	-	-	2.7	-	-
					BG + XY	+		-	-	-	-	-	-	-	3.8	-	-
[142]	-	-	990	M	BG + XY	-	35	-	-	-	13.8	9.0		9.8	8.8	8.6	-
					-	+		-	-	-	10.8	5.6		23.0	7.8	7.4	-
					BG + XY	+		-	-	-	13.8	10.1		26.2	13.2	12.9	-
[13]	H	N	962, 917 (for N)	M	BG + XY	-	21	-	-	-	1.9	0.51	-	-	3.4	3.6	-
	HL	W						-	-	-	4.1	7.4	-	-	9.4	9.6	-

^1^ Hulled (H) or hull-less (HL); ^2^ Normal (N), high amylose (HA) or waxy (W); ^3^ Mash (M) or pellets (P); ^4^ β-glucanase (BG) or xylanase (XY); ^5^ WG, weight gain; FI, feed intake; F/G, feed per gain; ^6^ N, nitrogen; P, Phosphorus; AME, apparent metabolisable energy; AMEn, N-corrected AME; ^7^ Growth performance determined at d 13. Nutrient utilisation and viscosity values determined at d 18.

### 5.3. Feed Processing

#### 5.3.1. Particle Size

Cereal grains are ground during feed manufacture to modify their physical characteristics by reducing the particle size. Grinding of whole grains can be categorised into three classes as fine, medium, and coarse according to the screen size in a hammer mill or distance between the rollers in a roller mill [199]. Morel and Cottam [200] achieved three different sizes of barley particles by grinding barley through a hammer mill (7.0, 4.0 and 1.0 mm sieve openings for coarse, medium and fine grinds, respectively) and, reported average particle sizes of 1100 μm for coarse, 785 μm for medium, and 434 μm for fine grinds in barley-based pig diets. Perera et al. [149] ground normal starch hulled barley in a hammer mill to pass 2.0 and 8.0 mm sieve sizes and reported the average particle sizes of 648 and 1249 μm for fine and coarse grinds, respectively. 

The particle size of a milled product can be influenced by grain type and, grinding different grains in the same mill under similar conditions can result in different particle sizes due mainly to the variations in endosperm hardness [199]. In accordance, it has been speculated that the variation in barley kernel hardness is responsible for the differences in particle size distribution observed between hard and soft barley lines [201]. Nair et al. [201], compared the microscopic images of endosperm from hard and soft-hulled spring barley lines and reported thicker endosperm cell walls in hard barley lines. Moreover, [162] reported that both β-glucan and arabinoxylan in barley endosperm are positively correlated with kernel hardness. It is therefore reasonable to speculate that barley NSP may indirectly influence the particle size distribution in different barley types. 

The grinding extent of barley has been compared with other physical manipulations such as WB feeding, pelleting and grit supplementation [202,203] in poultry diets. Comparing the effect of different particle sizes of barley on broiler performance and nutrient digestibility to determine the optimum barley particle size in pelleted broiler diets, Perera et al. [149] reported that coarsely ground barley improved F/G by 2.1 points and AMEn by 0.10 MJ/kg. Moreover, coarse grinding increased the CAID of DM, N and fat by 3.1, 3.2 and 4.3%, respectively, compared to those fed fine-barley diets. Tari et al. [151] evaluated the influence of the method of barley inclusion (fine, coarse and WB) in a wheat-based diet and reported that birds fed coarse and WB diets had higher WG than those fed fine barley diets by an average of 36.5 g/bird at 21 d of age, attributed to the increases in digestibility of nutrients. These findings are suggestive of the potential to enhance the feeding value of barley in poultry diets through strategic manipulation of feed processing practices. 

#### 5.3.2. Feed Form

Supporting the established fact that pelleting enhances the economics of production by improving the growth and feed efficiency responses in broilers [204], pelleted barley-based diets have been also reported to improve growth performance over mash diets [198,205,206]. Al Bustany [206] reported that pelleting a barley-based diet (500 g barley and 200 g maize/kg of diet) enhanced BW, FI and feed efficiency of 21-d broilers by 36 g/bird, 40 g/bird and 6 points, respectively. Comparing barley-based diets (450 g/kg) fed as either unprocessed mash or ground pellets, Lamp et al. [198] reported that broilers (d 21) fed ground pellets resulted in greater WG (611 vs. 665 g/bird) and FI (879 vs. 954 g/bird) compared to the birds fed unprocessed mash diets. Feed efficiency, however, was not affected by the feed form. Ankrah et al. [110] reported no effect of pelleting of either normal or waxy starch hull-less barley on growth performance of 21-d old broilers. The discrepancies in the extent of growth performance responses of broilers fed barley-based diets to feed form are presumably driven by, inter alia, the variability in barley types and different conditions employed during the pelleting process.

It is recognised that physical stress of pelleting can break the cell walls releasing the encapsulated nutrients leading to a greater accessibility by digestive enzymes [204]. In agreement, Ankrah et al. [110] reported a 17% increase in starch digestibility in broilers fed barley-based diets due to pelleting. Conversely, no effect of feed from on AA digestibility [138], AMEn, DM or N retention [207] in broilers fed barley-based diets has been reported. 

Pelleting can increase soluble carbohydrate concentrations or change the molecular weight of S.NSP, leading to an increase in digesta viscosity [204]. Al Bustany [206] reported that pelleting a barley-based diet increased the occurrence of sticky droppings of broilers (d 1−7) by 223%, due probably to an increase in digesta viscosity. However, Lamp et al. [198] reported no difference in digesta viscosity in broilers fed barley-based diets either as unprocessed mash or ground pellets. Ankrah et al. [110] reported 45% reduction in viscosity of barley after pelleting, an observation that was attributed to the shearing effect of the pelleting process that facilitated β-glucan degradation. 

#### 5.3.3. Heat Processing

Different heat processing methods such as autoclaving [71,208,209], steam-cooking [106], steam-conditioning [198,206], expansion, micronisation [210] and extrusion [211] have been evaluated to enhance the feeding value of barley in poultry diets. Heat processing is believed to disrupt the cell structures and to release the encapsulated nutrients [106,210] facilitating the nutrient utilisation. However, thermal processing can increase the solubilisation of NSP in cereal grains [212], leading to higher viscosity in both feed and intestinal contents [210,213] with an exacerbated effect on diets based on viscous grains such as barley [85]. In addition, other common drawbacks of employing extreme heat treatments such as; formation of resistant starch [214,215], degradation of heat-labile AA [216], inactivation of synthetic vitamins [217] and exogenous enzymes [108,218] also apply to cereal-based diets. 

Impaired WG, feed efficiency and nutrient utilisation in birds fed autoclaved barley (121 °C for 20 min) compared to those fed non-treated barley have been reported in the literature [208,209]. According to Vranjes and Wenk [211], feeding extruded barley deteriorated F/G and dietary AME in broilers by 3.9 points and 0.82 MJ/kg, respectively. These researchers reported an increased viscosity of barley extract (1.3 vs. 3.7 cP) due to an increase in concentrations of S.NSP (28.4 vs. 36.2 g/kg) induced by extrusion (120−130 °C for 20 s). In contrast, applying comparatively mild conditions, Viveros et al. [71] demonstrated that autoclaving (70 and 90 °C for 10 min) of enzyme-supplemented barley-based diet improved the growth performance of young broilers compared to the unprocessed control diet. 

Gracia et al. [106], using broiler starters (d 1−21), evaluated steam-cooked barley grains in mash diets, without or with a multi-component enzyme. An interaction between steam cooking (99 ± 2 °C for 50 min) and enzyme addition was reported for intestinal digesta viscosity with a greater response to enzyme in steam-cooked barley. Broilers fed steam-cooked barley grew faster than broilers fed unprocessed barley only up to 8 d of age. The F/G of broilers fed steam-cooked barley at 21 d of age was deteriorated by 8 points due likely to the 82% increase in intestinal digesta viscosity of broilers due to steam-cooking. 

García et al. [210] reported that heat processing of barley increased the intestinal digesta viscosity at 7-d of age resulting in viscosity of 270, 121, and 89 cP for micronised, expanded, and non-processed barley, respectively. The effect of heat processing on intestinal digesta viscosity, however, disappeared at d 42. Micronisation and expansion, however, improved the NSP digestibility by 14.5 and 27.8%, respectively, confirming the heat induced NSP solubilisation. Birds fed micronised barley gained less weight and had poorer F/G than broilers fed expanded barley, suggesting that micronisation might have a more severe impact on barley compared to the mild heating by expansion. Moreover, benefits of heat processing on barley seemed to be limited to first week of age of broilers [71,106,210].

While most studies have compared different methods of heat processing, studies evaluating the optimum pelleting conditions for barley-based diets are limited [107,108,218]. Inborr and Bedford [218] reported that WG and feed efficiency in broilers decreased following conditioning a barley-based diet at 95 °C compared to diets conditioned at 75 and 85 °C. Samarasinghe et al. [107] reported that conditioning temperature of 90 °C compared to 60 °C in a non-enzyme supplemented barley-maize-soy diet numerically impaired the WG, daily FI and F/G of broilers (d 7−21). Moreover, conditioning non-supplemented barley-maize-soy diet at 75 and 90 °C increased the dietary viscosity by 0.11 and 0.29 cP, respectively, compared to the diet conditioned at 60 °C. Perera et al. [108] evaluated the effect of three steam-conditioning temperatures (60, 74 and 88 °C) on normal starch hulled barley-based diets and reported impaired WG, feed efficiency, and ileal digestibility of N and starch in birds fed diets conditioned at 88 °C. Unlike in non-viscous grains, despite the more durable pellets obtained in diets conditioned at 88 °C, feed efficiency and nutrient utilisation were severely compromised, most likely due to the increased digesta viscosity, leading to the recommendation that normal starch hulled barley diets should be steam-conditioned up to 74 °C. 

Based on the limited available literature, it can be hypothesised that the conditions (heat, moisture and mechanical pressure) applied during the heat processing, rather than the heat processing method are of higher importance in barley-based diets. It, therefore, necessitates the determination of optimum conditions for each heat treatment, particularly pelleting process, used for manufacturing barley-based broiler diets. Recent findings by Perera et al. [108] showed that the response of viscous grains such as barley to increasing conditioning temperature differs from those of non-viscous grains and hence highlight the need to determine grain-specific optimum conditioning temperature. Moreover, thermal processing conditions can also interact with exogenous carbohydrases in barley-based diets, due to high temperature-induced viscosity increase and partial inactivation of enzymes during heat processing [106,218]. A better understanding of the interactions between exogenous enzymes and heat processing conditions, particularly on intestinal digesta viscosity and nutrient utilisation, is vital to minimise the viscosity related negative consequences and to facilitate increased use of barley in contemporary highly processed poultry diets.

#### 5.3.4. Whole Barley Feeding

Feeding whole grains has traditionally been a part of backyard poultry operations. The importance of whole grains in poultry nutrition has been recognised due to its benefits associated with a better developed and more functional gizzard. Moreover, whole grain feeding can lower the feed milling cost and enhance the gut integrity of broilers fed highly processed diets. Different methods of whole-grain feeding have been reported in the literature and extensively reviewed in Singh et al. [219]. 

Wheat is usually considered as the whole grain of choice, and barley is used as an alternative only when the cost or supply discourages the use of wheat [219]. Whole barley has been recognised less preferred in free-choice feeding method when chickens were offered alternatives [220]. Nevertheless, barley has been used in mixed feeding methods, for its greater impact on gizzard development compared to other whole grain types [221]. Whole barley has been investigated in maize—[221,222], wheat—[150,223] and sorghum- [223,224] based diets for the determination of optimum inclusion level and possible interactions with supplemental enzymes [203,223,225].

Reduced incidence of dilated proventriculus in response to WB has been evident [223,224], confirming barley potential for enhancing gut integrity. Even though enhanced gizzard development and functionality is the motivation for whole grain feeding, the effect of WB on gizzard development seemed to be inconsistent. While most studies [150,151,203,223,224] reported increased gizzard weight in response to replacing ground grain fractions with WB, Nahas and Lefrancois [222] reported no effect of WB inclusion on gizzard development. Furthermore, gizzard development response to WB can be confounded by the inclusion level, type, quality and hardness of the grain, age of birds, and whole grain feeding method. Nevertheless, with no difference in duodenal particle size distribution in broilers fed whole vs. ground barley-based diets, Svihus et al. [203] suggested a better grinding function by well-developed gizzards in broilers fed WB.

The effect of WB feeding on growth performance has been contradictory. Hetland et al. [150] reported that both WG and FI were impaired in broilers offered WB in wheat-based diets at inclusion levels of 125, 300 and 440 g/kg. Moss et al. [223] reported that post-pelleting inclusion of whole barley depressed WG by 74 g/bird, FI by 48 g/bird and FCR by 3.2 points compared to the ground barley fed birds at 28-d of age. In contrast, higher WG (744 vs. 693) and FI (1113 vs. 1037) in birds fed WB diets compared to those fed ground barley diets was reported by Svihus et al. [203]. The F/G, however, was not affected by the form of barley. Comparing different methods (fine, coarse and WB) of barley inclusion (283 g/kg) in a wheat-based diets, Tari et al. [151] reported that replacing fine barley with WB improved the WG by 42 g/bird and F/G by 2.7 points in 21-d old broilers. According to Nahas and Lefrancois [222], inclusion of WB (150 and 200 g/kg in the grower and finisher diets, respectively) in an enzyme supplemented maize-based diet, improved the WG and FI of broilers by 83 and 126 g/bird, respectively, compared to a non-supplemented maize-based diet without WB. Moreover, the inclusion of 150 g/kg WB in non-supplemented maize-based diet enhanced the F/G by 1.9 points, confirming the beneficial effects of WB inclusion in conventional maize-soybean broiler diets. However, Biggs and Parsons [221] reported similar WG in 21-d old broilers fed 100 and 200 g/kg WB to those fed a ground maize-soybean mash diet. The discrepancy in growth responses has resulted in varying WB inclusion levels being recommended for broiler diets. An inclusion of 300 g/kg [150] and 350 g/kg [226] of WB in broiler diets has been suggested without any adverse effects on bird performance. However, Nahas and Lefrancois [222] recommended a lower inclusion of up to 200 g/kg WB as an optimum level. 

The beneficial effects of WB feeding on gizzard development favourably influence the nutrient utilisation of birds. The enhanced starch digestibility (0.96 vs. 0.92) reported by Hetland et al. [150] in response to replacing ground barley with WB (440 g/kg) was attributed to 79% increase in relative gizzard weight (34 vs. 19 g/kg). Moss et al. [223] also reported a 1.05% increase in ileal starch digestibility parallel to the 21% increase in the relative gizzard weight in broilers fed 125 g/kg WB. According to Tari et al. [151], nutrient utilisation was benefited from 26% increase in relative gizzard weight in response to replacing fine or coarse barley with WB in a wheat-based diet.

Density of WB grain can prevent the proper mixing with concentrate portion of the mash feed and consequently induce segregation in the mixed feed. When WB is added post-pelleting, separation and floating of WB on the top of the feed bins can result in incomplete distribution. Moreover, WB from awned cultivars can be hazardous to young broilers resulting in perforation or impaction of the crop [219]. However, these limitations can probably be avoided by pre-pelleting inclusion of WB, with WB cracked and embedded in intact pellets. The possible interactions of WB feeding with supplemental enzymes [151,223], particularly carbohydrases that are commonly added to barley-based diets, merits further investigation.

### 5.4. Other Strategies to Enhance Barley Nutritional Value

Different pre-treatments such as soaking [181] and germination [225,227] have been investigated as possible strategies to enhance the nutritional value of barley for poultry. These treatments mainly focus on the activation of endogenous enzymes, mainly, β-glucanase [181]. Germinated barley was reported to have lower total and soluble β-glucan contents and digesta viscosity and, consequently improved growth performance and nutrient utilisation in broilers [225,227]. In comparison, the positive effect of soaking was not consistent and seemed to be dependent on the conditions (water temperature, time) employed during soaking [181,225].

Beyond the aim of sterilising the feed ingredients, gamma irradiation has been evaluated in barley grains prior to dietary inclusion to induce depolymerisation of β-glucan and consequent reduction in viscosity [208,209]. A 63% reduction in viscosity of a β-glucan solution in response to gamma irradiation was reported by Classen et al. [208]. When fed to broilers (d 1−21), irradiated hull-less barley improved WG and fat absorption compared to the non-treated barley [208]. Deteriorated growth performance and nutrient utilisation of broilers fed autoclaved barley was restored by subsequent irradiation of autoclaved barley [209]. Comparing two barley types subjected to gamma irradiation, Al-Kaisey et al. [228] reported a gradual decrease in viscosity of barley extract in response to increasing dose of gamma irradiation. However, the magnitude of the reduction in extract viscosity to irradiation dose was dependent on the barley type. The reduction in barley extract viscosity was attributed to depolymerisation of β-glucans, leading to lower β-glucan content and viscosity. In contrast, Campbell et al. [209] reported an increased soluble β-glucan content in barley in response to increasing levels of irradiation. Despite the higher soluble β-glucan content, these researchers reported a decline in barley extract viscosity, due probably to an irradiation-induced reduction in molecular size.

Fermentation of barley with *Lactobacillus*, *Bacillus* [229] and fungus [230] has been evaluated to improve the feeding value of barley for broilers. Kim and Kang [229] reported improved WG in broilers (d 1−35) fed fermented barley compared to those fed non-treated barley. Feed intake and F/G, however, were unaffected by the fermentation. Fermentation of barley with *Rhizopus oligosporus* improved WG over broilers fed untreated barley [230]. According to Yaşar et al. [231], inclusion of barley (400 g/kg in broiler starter diet and 450 g/kg in finisher diet), fermented by a solid-state fermentation process, improved the FI and F/G at d 21 and d 42 and, WG at d 42. These improvements were attributed to the reduction of digesta viscosity in birds compared to those fed unfermented barley. 

Application of biotechnology has also been investigated. Von Wettstein et al. [232] compared a non-supplemented barley-based diet and a diet containing transgenic barley grain (39 g/kg) containing (1,3−1,4)-β-glucanase and reported superior WG, FI and feed efficiency for broilers fed diets with transgenic barley. Moreover, increasing dietary inclusion of transgenic barley grain containing (1,3−1,4)-β-glucanase in barley-based diets up to 25.5 g/kg showed similar growth performance to that of broilers fed maize-based diets. 

However, most of these strategies are not economically attractive and their large-scale applications have been proven to be logistically difficult due to high cost and labour. Comparatively, supplementation of exogenous enzymes remains the most attractive approach because of its’ easy practice and lesser variability in response. A combination of compatible measures would facilitate each other mechanism enabling maximum efficacy in improving the feeding value of barley for poultry diets.

## 6. Conclusions

The fact that the nutritive value of barley for poultry is determined not only by the chemical and physical properties but also by the interactions of the nutrient and anti-nutrient components highlights the need for the application of grain-specific metabolisable energy and digestible nutrients, in particular AA, in formulating barley-based diets. Understanding the effects of feed processing and supplemental enzymes in barley-based diets enables nutritionists to manipulate conditions to minimise the inherent variability of barley grains and, consequently, increase inclusion of barley in broiler diets. Moreover, considering the variability of barley grain, processing conditions, enzyme supplementation and optimal inclusion should be tailored to the specific barley cultivar in use.

## Figures and Tables

**Table 1 animals-12-02525-t001:** Comparison of starch in maize, wheat, hulled barley and hull-less barley types (g/kg, DM basis).

Reference	Grain Type	Hull/ Hull-Less	Starch Type	n ^1^	Starch	Amylose ^2^	Amylopectin ^2^
[5]	Barley	Hull-less	Normal	2	642	158 (25)	483 (75)
			Normal (CG ^3^)	2	605	171 (28)	433 (72)
			High amylose	2	563	243 (43)	320 (57)
			Waxy	2	622	33 (5.0)	589 (95)
			Waxy (CG)	1	582	27 (5.0)	555 (95)
			Zero amylose waxy	1	585	0 (0)	585 (100)
	Maize ^4^		Normal	1	-	- (25)	- (75)
			Waxy	1	-	- (1.0)	- (99)
[32]	Wheat		Normal	1	605	163 (27)	442 (73)
			Waxy	1	563	18 (3.0)	545 (97)
	Maize ^4^		Normal	1	-	- (21)	- (79)
			Waxy	1	-	- (3.0)	- (97)
[33]	Barley	Hull-less	Normal	2	616	248 (40)	368 (60)
			High amylose	2	537	416 (77)	121 (23)
			Waxy	2	561	51 (9.0)	510 (91)
			Zero amylose waxy	2	533	0 (0.0)	533 (100)
[28]	Barley	Hulled	Normal	28	588	147 (25)	441 (75)
		Hull-less	Normal	6	609	152 (25)	457 (75)
		Hulled	Waxy	1	552	44 (8.0)	508 (92)
		Hull-less	Waxy	3	582	29 (5.0)	553 (95)
		Hull-less	High amylose	1	535	193 (36)	342 (64)
[34]	Barley	Hulled	Normal	1	598	168 (28)	430 (72)
		Hull-less	Normal	1	655	164 (25)	491 (75)
		Hull-less	Waxy	2	614	37 (6.0)	577 (94)
[13]	Wheat	Hulled	Normal	1	537	229 (43)	308 (57)
	Barley	Hulled	Normal	1	610	267 (44)	343 (56)
		Hull-less	Waxy	1	554	77.2 (14)	477 (86)

^1^ Number of analysed grain types; ^2^ Values in the parenthesis are amylose or amylopectin as a percentage of starch content; ^3^ CG, compound starch granules that exist in clusters of individual granules; ^4^ Total starch content was not reported.

**Table 2 animals-12-02525-t002:** Crude protein and amino acid (AA) composition of barley, maize and wheat (g/kg, DM).

Reference	[44]	[34]				[45]			[46]		[27]			[13]		
Grain Type	Wheat	Barley				Barley	Maize	Wheat	Wheat	Barley	Barley	Maize	Wheat	Wheat	Barley	
Starch Type		Normal		Waxy											Normal	Waxy
H/HL ^1^		H	HL	HL											H	HL
n ^2^	16	1	1	1	1	1	7	7	6	7	21	27	29	1	1	1
DM	879−889	890	899	894	903	896	895	898	940	921	882	903	877	892	893	907
CP	120	116	104	105	137	94.9	89.4	103	162	143	123	93.5	137	141	101	133
Indispensable AA																
Arginine	5.8	5.55	4.91	4.08	6.39	5.47	4.44	5.19	7.6	6.8	5.99	4.33	6.56	6.79	5.28	6.44
Histidine	2.9	3.11	2.58	2.3	3.45	2.57	2.71	2.85	3.8	3	2.9	2.87	3.47	3.46	2.35	2.82
Isoleucine	4.2	4.18	3.89	3.54	5.24	3.79	3.59	4.15	5.3	4.8	3.85	3.07	4.25	4.94	3.69	4.87
Leucine	7.6	8.15	7.24	6.47	10.1	7.7	12.1	7.77	10.5	9.9	8.3	11.78	9.14	9.82	7.02	8.99
Lysine	3.4	4.06	3.43	3.07	5.23	4.02	2.83	3.44	4.4	4.9	4.29	2.79	3.73	3.95	3.84	4.55
Methionine	1.8	1.85	1.69	1.66	1.89	1.45	1.63	1.45	2.5	2.4	1.93	1.93	2.01	2.52	2.16	2.23
Phenylalanine	5.1	6.56	5.13	4.51	8.16	5.36	4.77	5.08	7.4	7.6	6.3	4.63	6.37	6.99	5.13	7.31
Threonine	3.3	3.77	3.55	3.1	4.68	3.46	3.83	3.47	4.5	4.7	4.17	3.41	3.92	4.14	3.67	4.18
Valine	5.2	5.95	5.46	4.88	7.08	5.36	4.83	5.02	6.6	7	5.44	4.2	5.26	6.49	5.54	6.82
Tryptophan	-	-	-	-	-	1.23	0.46	0.54	-	-	1.51	0.7	1.58			
Dispensable AA																
Alanine	4.2	4.54	4.12	3.69	5.79	4.58	7.39	4.23	5.5	5.5	4.82	7.38	4.71	4.99	4.28	4.98
Aspartic acid	6	7.73	6.72	6.37	10.9	6.36	6.37	5.93	8	8.1	7.11	6.26	6.84	7.46	6.82	8.09
Cysteine ^3^	2.6	2.33	2.26	2.21	2.41	-	-	-	3.5	2.8	2.57	2.09	3.03	3.50	2.65	3.00
Glutamic acid	31.4	31.8	27.5	24.2	37.9	25.2	18.2	31	46.5	35.8	29.9	17.4	40.4	45.1	23.6	34.4
Glycine ^3^	4.8	4.62	4.02	3.56	5.54	4.35	3.83	4.8	6.5	5.4	4.74	3.47	5.53	5.95	4.38	4.99
Proline	11	14.2	11.4	10.37	18.3	-	-	-	16.4	15.9	15.62	9.82	15.76	15.2	10.6	16.1
Serine	5.5	4.53	4.26	3.63	5.25	4.91	4.64	5.86	7.2	6.1	5.4	4.74	6.67	7.10	4.50	5.23
Tyrosine	-	-	-	-	-	3.01	3	2.54	-	-	3.47	3.46	3.66	4.68	3.41	4.36

^1^ H, hulled; HL, hull-less; ^2^ Number of analysed grain types; ^3^ Semi-indispensable AA for poultry.

**Table 3 animals-12-02525-t003:** The type and content of non-starch polysaccharides in barley, maize and wheat (g/kg, DM basis).

Reference	Grain Type	n ^1^	Starch Type ^2^	H/ HL ^3^		NSP ^5^										Proportion of Total NSP (%)
						AX	A	X	BG	CE	M	GA	U	GL	Total	
[49]	Wheat	-	- - -	- - -	S ^4^	18	-	-	4.0	-	t ^8^	2.0	t	-	24	21
					I ^4^	63	-	-	4.0	20	t	1.0	2.0	-	90	79
	Barley ^6^	-			S	8.0	-	-	36	-	t	1.0	t	-	45	27
					I	71	-	-	7.0	39	2.0	1.0	2.0	-	122	73
	Maize	-			S	1.0	-	-	t	-	t	t	t	-	1.0	1.0
					I	51	-	-	-	20	2.0	6.0	t	-	80	99
[50]	Wheat	16	-	-	S	-	10	7.0	-	-	0.4	1.8	-	3.0	23	18
					I	-	41	25	-	-	1.3	1.4	-	34	103	82
[47] ^7^	Barley	1	N	H	S	77	2.4	3.2	22	40	0.7	0.7	1.5	32	40	17
					I		21	50	25		3.6	2.0	2.9	55	200	83
		1	HA	HL	S	90	4	5.6	26	47	1.4	0.8	1.7	49	63	20
					I		23	57	43		6.7	2.3	3.4	67	249	80
		1	W	H	S	75	3.3	4.6	31	35	0.9	0.8	2.1	46	58	23
					I		22	45	30		3.7	2.0	3.1	50	191	77
		1	N	H	S	83	2.6	3.2	15	42	0.8	0.7	1.1	21	29	13
					I		23	55	13		6.7	2.1	3.5	49	194	87
		1	N	HL	S	52	3.5	4.9	24	19	1	1.1	1.5	32	44	26
					I		17	27	22		3.9	1.6	1.9	33	125	74
		1	HA	HL	S	57	4.5	6.6	26	16	1.4	0.8	1.7	48	63	28
					I		18	28	48		4.7	1.5	1.9	42	160	72
		1	W	HL	S	48	2.8	3.6	30	14	0.9	0.7	1.9	37	46	27
					I		18	24	26		4.2	1.8	1.7	33	123	73
		1	W	HL	S	120	7.8	13	12	41	3.7	1.8	2.4	123	152	30
					I		38	61	137		10	2.9	3.3	67	360	70
[28]	Barley	28	N	H	S	13.7	-	-	-	127	-	-	-	-	106	31
					I	116	-	-	-		-	-	-	-	232	69
		1	W	H	S	15.5	-	-	-	177	-	-	-	-	184	45
					I	109	-	-	-		-	-	-	-	223	55
		6	N	HL	S	22.6	-	-	-	92	-	-	-	-	125	49
					I	66.1	-	-	-		-	-	-	-	128	51
		3	W	HL	S	24.1	-	-	-	127	-	-	-	-	200	64
					I	62.9	-	-	-		-	-	-	-	114	36
		1	HA	HL	S	20.5	-	-	-	140	-	-	-	-	222	64
					I	60.8	-	-	-		-	-	-	-	125	36

^1^ n, number of analysed samples; ^2^ N, normal; HA, high amylose; W, Waxy; ^3^ H, hulled; HL, hull-less; ^4^ S, soluble; I, insoluble; ^5^ AX, arabinoxylan; A, arabinose; X, xylose; BG, β-glucan; CE, cellulose; M, mannose; GA, galactose; U, uronic acid; GL, glucose; ^6^ [51]; ^7^ Total insoluble NSP = The sum of insoluble A, X, BG, MA, GAL, UA, GLU and total CEL; ^8^ t, Trace amounts.

**Table 4 animals-12-02525-t004:** Mineral composition of maize, barley and wheat grains (g/kg, DM basis).

Reference	[90]				[91]		[27]			[13]		
Grain Type	Barley		Maize		Barley		Barley	Maize	Wheat	Wheat	Hulled Barley	Hull-Less Barley
	Wild ^1^	LP ^2^	Wild	LP	Wild	LP						
n ^3^	1	1	1	1	1	1	21	27	29	1	1	1
Calcium	0.6	0.6	0.02	0.03	0.47	0.49	0.59	0.04	0.4	0.35	0.39	0.36
Phosphorus (P)	4.1	3.3	3.2	3.2	3.63	3.52	4.3	3.17	3.67	4.26	3.25	3.86
Phytate P	2.3	1.1	2.2	0.9	2.38	0.05	2.81	2.26	1.92	2.22	1.32	1.79
Non-phytate P	1.8	2.2	1	2.3	1.25	3.47	1.49	0.91	1.75	2.04	1.93	2.07
Magnesium	1.3	1.3	1.3	1.2	1.2	1.2	1.63	1.45	1.56	1.45	1.28	1.39
Potassium	-	-	-	-	-	-	5.53	3.96	4.33	4.93	4.25	5.62
Sodium	-	-	-	-	-	-	0.05	0.003	0.005	<0.06	0.20	0.10
Iron	-	-	-	-	0.062	0.071	0.04	0.02	0.04	0.06	0.06	0.06
Chloride	-	-	-	-	-	-	-	-	-	0.71	1.31	1.27
Manganese	0.017	0.02	0.007	0.007	0.016	0.015	0.015	0.005	0.032	-	-	-
Zinc	0.030	0.04	0.010	0.010	0.023	0.024	0.024	0.021	0.022	-	-	-
Copper	0.009	0.01	0.006	0.006	0.003	0.004	0.005	0.002	0.004	-	-	-

^1^ Wild-type barley with normal phytate P content; ^2^ Low-phytate; ^3^ Number of analysed samples.

**Table 5 animals-12-02525-t005:** Comparison of different cereal types for intestinal digesta viscosity of broilers.

Reference	Grain	Inclusion Level (g/kg of Diet) (g/kg Diet)	Sampling Point	Major NSP ^1^ (g/kg)	Bird Age (d)	Viscosity (cP)
[111]	Maize	452	Small intestine	Soluble BG: 0.2	14	1.7
	Barley	698		Soluble BG: 17.2		2.4
[98]	Maize	600	PSI ^2^	-	22	1.0
	Low viscosity barley			Total BG: 32.3		13
	High viscosity barley			Total BG: 38.7		29
[110]	Hull-less normal starch barley	610	PSI ^2^	Total BG: 60	21	178
			DSI ^2^			353
	Hull-less waxy starch barley		PSI	Total BG: 73		376
			DSI			440
[112]	Triticale	686/719 ^3^	Ileum	Soluble AX:12.3	35	6.0
	Rye	621/652 ^3^		Soluble AX: 27.3		140
	Wheat	745/740 ^3^		Soluble AX: 10.6		3.0
[101]	Barley	600	Ileum	-	28	3.2
	Sorghum	623		-		2.2
	Wheat			-		7.3
	Maize			-		2.4
[109]	Wheat	657	Foregut ^4^	-	Average value of 20, 25, 30, 35	2.7
			Hindgut ^4^	-		8.0
	Barley	660	Foregut	-		21
			Hindgut	-		28
[80,81]	Wheat	629	Jejunum	Total BG: 7.74	21	4.99
	Hulled normal starch barley	565		Total BG: 38.5		2.81
	Hull-less waxy starch barley	314/260 ^5^		Total BG: 68.6		3.51

^1^ NSP, non-starch polysaccharides; BG, β-glucan; AX, Arabinoxylan; ^2^ PSI, proximal small intestine (from gizzard to Meckel’s diverticulum); DSI, distal small intestine (from Meckel’s diverticulum to the ileo-caecal junction); ^3^ Starter (1−14 d)/finisher (15−35 d) diet composition; ^4^ Foregut, duodenum to Meckel’s diverticulum; Hindgut, from Meckel’s diverticulum to the ileo-caecal junction; ^5^ Wheat/Hull-less waxy starch barley.

**Table 6 animals-12-02525-t006:** Comparison of apparent metabolisable energy (AME; MJ/kg DM) and nitrogen-corrected AME (AMEn; MJ/kg DM) of different cereal grains for broilers.

Reference	Grain Type	AME	AMEn
[125]	Pearled rice	17.36	
	Maize	15.83	
	Sorghum	15.77	
	Wheat	14.32	
	Triticale	13.83	
	Barley	11.92	
	Rye	11.34	
[116]	Maize	14.01	
	Hull-less barley	11.12	
	Hulled barley	10.05	
[34]	Hull-less normal starch barley		12.97
	Hulled normal starch barley		12.72
	Hull-less waxy starch barley		11.23
[126]	Sorghum	15.0	
	Barley	12.5	
[119]	Maize Barley	10.75	
	Wheat	10.74	
	Sorghum	10.64	
	Barley	9.91	
[13]	Hull-less waxy starch barley	11.38	11.11
	Hulled normal starch barley	13.90	13.63
	Wheat	14.71	14.40
[127]	Maize	14.64	14.39
	Sorghum	14.00	13.74
	Wheat	11.10	10.78
	Barley	10.24	9.92

**Table 7 animals-12-02525-t007:** Comparison of standardised ileal digestibility of amino acids (AA) in different cereal grains.

Reference	[46]		[13]			[137]						[138]			
Age of the Birds (d)	21		21			14			28			24			
Grain Type ^1^	W	B	W	B ^2^		W	T	B	W	T	B	M	S	W	B
				NSH	WSHL										
Crude protein	0.872	0.797	0.849	0.788	0.747	-	-	-	-	-	-	0.829	0.808	0.797	0.711
Indispensable AA															
Arginine	0.852	0.804	0.838	0.786	0.748	0.89	0.79	0.77	0.87	0.88	0.82	0.873	0.841	0.746	0.715
Histidine	0.870	0.807	0.896	0.797	0.750	0.90	0.86	0.76	0.89	0.93	0.91	0.841	0.737	0.775	0.714
Isoleucine	0.904	0.839	0.868	0.791	0.745	0.91	0.85	0.81	0.89	0.93	0.92	0.825	0.809	0.769	0.684
Leucine	0.905	0.848	0.885	0.811	0.760	0.92	0.88	0.83	0.88	0.96	0.88	0.898	0.843	0.805	0.736
Lysine	0.837	0.805	0.832	0.749	0.707	0.85	0.79	0.78	0.82	0.82	0.82	0.767	0.758	0.635	0.639
Methionine	0.914	0.883	0.914	0.846	0.757	0.92	0.85	0.76	0.90	0.90	0.82	0.890	0.846	0.813	0.760
Phenylalanine	0.938	0.909	0.897	0.814	0.780	0.9	0.84	0.8	0.89	0.91	0.85	-	-	-	-
Threonine	0.854	0.806	0.784	0.708	0.671	0.79	0.81	0.75	0.79	0.88	0.87	0.809	0.794	0.727	0.701
Valine	0.877	0.825	0.832	0.786	0.740	0.86	0.86	0.79	0.88	0.93	0.86	0.833	0.808	0.755	0.722
Tryptophan	-	-	-	-	-	0.90	0.90	0.86	0.89	0.95	0.92	0.719	0.799	0.747	0.667
Dispensable AA															
Alanine	0.838	0.781	0.826	0.740	0.699	0.83	0.82	0.76	0.79	0.91	0.79	0.878	0.843	0.692	0.671
Aspartic acid	0.838	0.781	0.820	0.753	0.726	0.87	0.84	0.75	0.76	0.96	0.81	0.818	0.814	0.682	0.674
Cysteine	0.908	0.839	0.816	0.819	0.763	0.88	0.82	0.75	0.84	0.82	0.79	0.857	0.781	0.862	0.815
Glutamic acid	0.966	0.876	0.957	0.873	0.819	0.96	0.92	0.85	0.95	0.93	0.91	0.895	0.847	0.914	0.804
Glycine	0.841	0.767	0.818	0.722	0.682	0.83	0.82	0.72	0.79	0.84	0.85	0.745	0.713	0.731	0.652
Proline	0.954	0.866	0.925	0.856	0.811	0.95	0.89	0.86	0.96	0.94	0.91	0.864	0.797	0.912	0.808
Serine	0.891	0.822	0.828	0.738	0.693	0.87	0.83	0.76	0.86	0.91	0.83	0.858	0.831	0.824	0.736
Tyrosine	-	-	0.889	0.795	0.737	0.89	0.85	0.76	0.91	0.95	0.87	-	-	-	-

^1^ W, wheat; B, barley; T, triticale; M, maize; S, sorghum; ^2^ NSH, normal starch hulled barley; WSHL, waxy starch hull-less barley.

**Table 8 animals-12-02525-t008:** Ileal starch digestibility of different grain types fed to broilers.

Reference	Grain Type	H/HL ^1^	Starch Type	Total β-Glucans (g/kg)	Starch Digestibility Coefficient
[35]	Barley	H	Normal	31 ^2^	0.91
	Barley	H	Waxy	40 ^2^	0.87
	Barley	H	High Amylose	39 ^2^	0.89
[67]	Wheat	-	-	-	0.79
	Barley	-	-	-	0.96
	Oat	-	-	-	0.99
[144]	Wheat	-	-	-	0.944
	Maize	-	-	-	0.970
	Barley	-	-	-	0.981
	Sorghum	-	-	-	0.953
[34]	Barley	H	Normal	50	0.804
	Barley	HL	Normal	40	0.837
	Barley	HL	Waxy	64	0.587
[101]	Maize	-	-	-	0.95
	Wheat	-	-	-	0.97
	Sorghum	-	-	-	0.93
	Barley	-	-	-	0.93
[13]	Wheat	-	-	7.74	0.987
	Barley	H	Normal	38.5	0870
		HL	Waxy	68.6	0.987
[127]	Maize	-	-	-	0.991
	Sorghum	-	-	-	0.967
	Wheat	-	-	-	0.973
	Barley	-	-	-	0.943

^1^ Hulled (H) or hull-less (HL); ^2^ Soluble β-glucans for normal, waxy and high amylose barley types were 14, 22 and 12 g/kg, respectively.

**Table 9 animals-12-02525-t009:** Comparison of studies evaluating barley inclusion in broiler diets.

Reference	Barley Type		Replaced or Compared with	Inclusion Levels of Barley (g/kg Diet)	Method of Determination				Diets are Balanced for	
	Starch Type	H/HL/ DH ^1^			Weight-to-Weight Basis	Grain Chemical Composition	Table Values	Digestible AA ^2^	Energy	Protein
[128]	Waxy	H	Wheat	Starter, 0, 272, 408 and 544; Grower, 0, 323, 485, 646	Yes	No	No	No	No	No
	Normal	H			Yes	No	No	No	No	No
[118]	-	H	Wheat	0, 350, 700	Yes	No	No	No	No	No
	-	HL	Wheat	0, 350, 700	Yes	No	No	No	No	Yes
[104]	-	-	Maize	300, 400, 500, 600	Yes	Yes	No	No	No	No
[105]	-	-	Maize	0, 70, 140, 278, 557	No	No	Yes	No	Yes	Yes
				0, 79, 157, 314, 627				No	Yes	Yes
[113]	-	DH	Maize	0, 400, 800	Yes	Yes	No	No	Yes	Yes
[101]	-	-	Wheat, Maize, Sorghum	600.2	No	No	Yes	No	Yes	Yes
[119]	-	-	Wheat, Maize, Sorghum	Starter diet, 652; Finisher diet, 669	No	No	Yes	No	Yes	Yes
[80]	Normal	H	Wheat	0, 141, 283, 424, 565	No	Yes	No	Yes	Yes	Yes
[81]	Waxy	HL	Wheat	0, 65, 130, 195, 260	No	Yes	No	Yes	Yes	Yes

^1^ Hulled (H), hull-less (HL), de-hulled (DH); ^2^ Using digestible amino acid contents.

## Data Availability

All available data incorporated in the manuscript.

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
