# Peer review of "Barley, an Undervalued Cereal for Poultry Diets: Limitations and Opportunities"

_animals, 2022, doi:10.3390/ani12192525_

Round 1

Reviewer 1 Report

·         The present topic is very important in improving the sustainable poultry nutrition in the global village. Alternative cereals to maize are needed as maize faces competing uses in the ever-increasing world population dispensation. The discourse on nutritive value of barley as potential substitute for maize in livestock nutrition   must be dealt with in detail. Dismal documentations have been done on potential of barley as feed ingredients for animals under captivity and research studies on barley are somewhat scanty.

·         I have the following comments for consideration

·         It is important to include a section on anti-nutrition factors (ANF) and provide a detailed and lengthy discussion how ANF impact animal health.

·         I recommend another section of tabulated data on comparison of nutritive value of Barely and other cereals that have been used in livestock nutrition with documented results. It would also include the performance on animals.

·         Please add information on fermentation of Barley as way of improving digestibility, growth performance and palatability as well reducing ANF. Refer to studies conducted on Barley in fish and other animals.

·         I recommend that  you show results of the nutritive value of transgenic barley (after microbial or genetic engineering of the crop)

Author Response

 The comments by the reviewer are much appreciated. We have favourably considered the comments and the changes are highlighted in YELLOW in the amended version

It is important to include a section on anti-nutrition factors (ANF) and provide a detailed and lengthy discussion how ANF impact animal health.

Response: The suggestion is noted. However, the ANFs in barley and their negative effects on animals have been VERY extensively dealt – in relation to barley – throughout the text. As such, having a separate section on ANFs will not be relevant and also beyond the scope of this review.

I recommend another section of tabulated data on comparison of nutritive value of Barely and other cereals that have been used in livestock nutrition with documented results. It would also include the performance on animals.

Response: Comparative data with maize and other cereals are already extensively covered in the Tables and text throughout the original submission (for example, see Tables 1 to 9). We therefore believe that a separate section of tabulated data will be confusing.

Please add information on fermentation of Barley as way of improving digestibility, growth performance and palatability as well reducing ANF. Refer to studies conducted on Barley in fish and other animals.

Response: Suggested contents were added. Please refer to L1148-1156. New references added [220,221, 222]. See L1631-1636.

I recommend that you show results of the nutritive value of transgenic barley (after microbial or genetic engineering of the crop)

Response: Suggested contents were added. Please refer to L 1157-1162. . New reference added [223]. See L1637-1638.

Reviewer 2 Report

This is a fairly well written, comprehensive review of the challenges and potential for barley's use in poultry feeding. A few minor suggestions:

In section 3.1: Suggest adding a cross-sectional diagram of barley showing grain anatomy

In section 3.2.2: For readability, suggest adding a reference to Table 2 in the first paragraph. As currently laid out, there is no context for Table 2 until the page after Table 2.

Line 208: ...gut mortality...?

Line 219-221: Suggest rewriting sentence. Not overly clear.

Line 233-234: Suggest adding specificity related to "...modify gut physiology..." and "...interact with gut microflora..."

Line 286: Beginning of sentence appears to be missing?

Table 4: Need to add identifier for the numbers in brackets at the top. Those are references, but that needs to be indicated.

Overall, this is a well written review. I would suggest authors carefully re-read the manuscript looking for sentence structure and syntax issues.

Author Response

The comments by the reviewer much appreciated. We have favourably considered all the comments and the changes are highlighted in YELLOW in the amended version.

In section 3.1: Suggest adding a cross-sectional diagram of barley showing grain anatomy

Response: This will require copyright permission, which may take (based on previous experience) considerable time and delay the publication. We respectfully decline, partly because addition of this figure will not be of added value.

Section 3.2.2: For readability, suggest adding a reference to Table 2 in the first paragraph. As currently laid out, there is no context for Table 2 until the page after Table 2.                     Response: Good point.  Reference to Table 2 is now made to the first paragraph (L-157)

Line 208: ...gut mortality..

Response: Corrected.

Line 219-221: Suggest rewriting sentence. Not overly clear.

Response: Now edited to make it clearer.

Line 233-234: Suggest adding specificity related to "...modify gut physiology..." and "...interact with gut microflora..."

Response: Now edited with specified information (see L233-237)

Line 286: Beginning of sentence appears to be missing?

Response: Thanks for this pick-up. Revised.

Table 4: Need to add identifier for the numbers in brackets at the top. Those are references, but that needs to be indicated.

Response: Revised.

Overall, this is a well written review. I would suggest authors carefully re-read the manuscript looking for sentence structure and syntax issues.                                              

Response: we have carefully gone through the text and revised any syntax issues